# Confidence modulates the decodability of scene prediction during partially-observable maze exploration in humans

Risa Katayama [1✉], Wako Yoshida[2,3,6] & Shin Ishii[1,4,5,6]

Prediction ability often involves some degree of uncertainty—a key determinant of confidence. Here, we sought to assess whether predictions are decodable in partially-observable environments where one's state is uncertain, and whether this information is sensitive to confidence produced by such uncertainty. We used functional magnetic resonance imaging-based, partially-observable maze navigation tasks in which subjects predicted upcoming scenes and reported their confidence regarding these predictions. Using a multi-voxel pattern analysis, we successfully decoded both scene predictions and subjective confidence from activities in the localized parietal and prefrontal regions. We also assessed confidence in their beliefs about where they were in the maze. Importantly, prediction decodability varied according to subjective scene confidence in the superior parietal lobule and state confidence estimated by the behavioral model in the inferior parietal lobule. These results demonstrate that prediction in uncertain environments depends on the prefrontal-parietal network within which prediction and confidence interact.

[1] Graduate School of Informatics, Kyoto University, Kyoto, Kyoto 606-8501, Japan. [2] Nuffield Department of Clinical Neuroscience, University of Oxford, Oxford OX3 9DU, UK. [3] Department of Neural Computation for Decision-making, Advanced Telecommunications Research Institute International, Soraku-gun, Kyoto 619-0288, Japan. [4] Neural Information Analysis Laboratories, Advanced Telecommunications Research Institute International, Soraku-gun, Kyoto 619-0288, Japan. [5] International Research Center for Neurointelligence, The University of Tokyo, Bunkyo-ku, Tokyo 113-0033, Japan. [6]These authors contributed equally: Wako Yoshida, Shin Ishii. ✉email: katayama.risa.44n@st.kyoto-u.ac.jp

Animals, including humans, have difficulty in decision-making due to past, present, and future uncertainties related to their geographical surroundings. For instance, when navigating a complex environment, knowing which way to go is more challenging if you are uncertain about your initial location, especially in a setting in which many places look similar. Working out both where you are and where you are going involves accepting assumptions and using them to make predictions. These beliefs can then be updated based on whether your predictions are correct or not. Over time, this leads to varying levels of confidence about your whereabouts and ability to accurately predict future conditions.

Predictions are considered high-order functions since they incorporate varying degrees of uncertainty to facilitate effective decision-making. Additionally, these predictions are considered subjective because they may not be entirely rooted in objective reality. In fully observable decision-making contexts in which one's current state is certain, it has been shown that explicit predictions can be decoded from brain activity. This has been demonstrated in the context of spatial navigation[1,2] and perceptual decision-making tasks[3–5] but has not yet been determined in partially observable environments. Although previous studies of Bayesian modeling reproduced subjective beliefs in predictions during a human navigation task in a partially observable maze[6], no studies have specifically investigated the decodability of explicit predictions.

Assuming that confidence is a measure of introspection to neural representations, it would increase as the environment becomes more predictable, to the point that the uncertainty of the belief would be resolved. Likewise, the decodability of predictions would also increase as the predictions become more confidently refined[7]. Previous studies have consistently shown that anterior prefrontal cortical activity correlates with subjective confidence estimates[8–11]. It was also reported that information necessary for decision-making, such as value difference[9] and liking rating[10], is represented by brain activity in the same prefrontal region. In the context of multi-voxel pattern analysis (MVPA), confidence has been decoded from regions in the prefrontal and parietal cortices[12,13], and one parietal region was identified to encode decisions[12]. Although these studies have suggested that internal representations and their introspections are tightly coupled, uncertainty stems directly from the experimental stimuli. In reality, environmental uncertainty or unpredictability must be dynamically resolved by continuously incorporating new information into the decision-making process[14]. Therefore, confidence changes are associated with uncertainty resolution. However, the relationship between confidence and decodability for predictions has never been assessed under dynamic, lifelike conditions because of difficulties in monitoring and reproducing dynamically changing internal representations. Thus, we used Bayesian modeling, which was also used to assess neurophysiological activities during a rodent goal-reaching task[15], to study and reproduce subjects' predictions and confidence based on their behaviors.

In this study, we sought to examine the following: (i) whether explicit predictions of the upcoming scene can be decoded from brain activity, and, if so, where this information is localized, and (ii) whether subjective confidence in these predictions can be decoded, and how confidence interacts with the predictions themselves. Accordingly, we conducted functional magnetic resonance imaging (fMRI) scanning during a virtual maze navigation task in which subjects explored a previously learned maze consisting of four-walled rooms with either an open or a closed door on each wall (Fig. 1, Supplementary Fig. 1). Subjects were initially placed in an unknown room with an unknown orientation (state) and provided only 3D scenes of the rooms. When subjects explored the maze by selecting one of the open doors leading to the next room, they were occasionally asked to predict the 3D scene that would be seen in the next room and to report their confidence for that prediction. This task involved decision-making combined with uncertainty resolution stemming from the partial observability of the environment, that is, the uncertainty in our paradigm is a composite of the uncertainty about the observed information in the context of standard perceptual tasks and the uncertainty arising from memory limitations in pure memory tasks. We hypothesized that scene prediction in a partially observable environment would be encoded in the parietal and prefrontal cortices, where activity is frequently observed in spatial navigation and planning[16–19], while its confidence would be encoded in the anterior part of the prefrontal cortex[8,9]. However, it remains unclear whether prediction and confidence interact at the level of neural activity. We supposed that if the confidence modulates the neural representation encoding scene prediction, the decodability of the prediction would vary with the confidence level.

## Results
Thirty-three healthy subjects (aged 20–32 years; four females) performed the maze navigation task (Fig. 1). Six subjects were excluded from the analyses due to low scene prediction accuracies (27.1–31.5%, see also Supplementary Fig. 2). Missed trials (mean frequency ± standard deviation [SD]: 0.7 ± 1.1%), defined as trials in which subjects did not complete the upcoming scene prediction and confidence reporting within the allotted time (4.5 s), were also excluded from the analyses of individuals.

**Behavioral results.** The average prediction accuracy was 54.6 ± 14.4% (mean ± SD) for 27 subjects who were included in the behavioral analysis. Figure 2a represents an example of a subject's behavioral profile in the prediction trials of four consecutive games. The prediction accuracy was significantly higher when the confidence was high (confidence levels 3 and 4) compared to when it was low (confidence levels 1 and 2) (Fig. 2b, one-sided Wilcoxon signed-rank test, $p = 3.3 \times 10^{-6}$). Subjects were able to accurately assess their confidence even though they were required to do so *before* they chose a predicted scene.

The initial states varied for each game and remained unknown to the subjects. Therefore, to predict an upcoming scene, the subjects were required to infer the hidden current state from the history of actions and observed scenes (i.e., the task was a partially observable Markov decision process [MDP][20]). As such, the uncertainty about the current state could be resolved as subjects continued looking for the goal, and accordingly, the scene prediction accuracy and confidence would have likely increased. The scene prediction accuracy increased with every successive prediction trial within a single game (Fig. 2c, $r = 0.27$, $p = 1.3 \times 10^{-5}$), as did the confidence level (Fig. 2d, $r = 0.29$, $p = 2.6 \times 10^{-6}$). Furthermore, the prediction accuracy was significantly correlated with the number of recent consecutive correct prediction trials (Fig. 2e, $r = 0.34$, $p = 6.6 \times 10^{-5}$); however, prediction accuracy was not correlated with the number of consecutive incorrect prediction trials ($r = -0.08$, $p = 0.35$). These results lend support to our assumption that the subjects resolved state uncertainty and were able to infer their state more accurately with continued maze exploration.

We predicted that the subjects would make quicker choices when their confidence levels regarding the upcoming scene prediction were higher than when they were lower. Aligned rank transformation analyses of variance revealed that scene choice reaction times (RTs) were significantly shorter in the high-confidence trials than in the low-confidence trials (Fig. 2f,

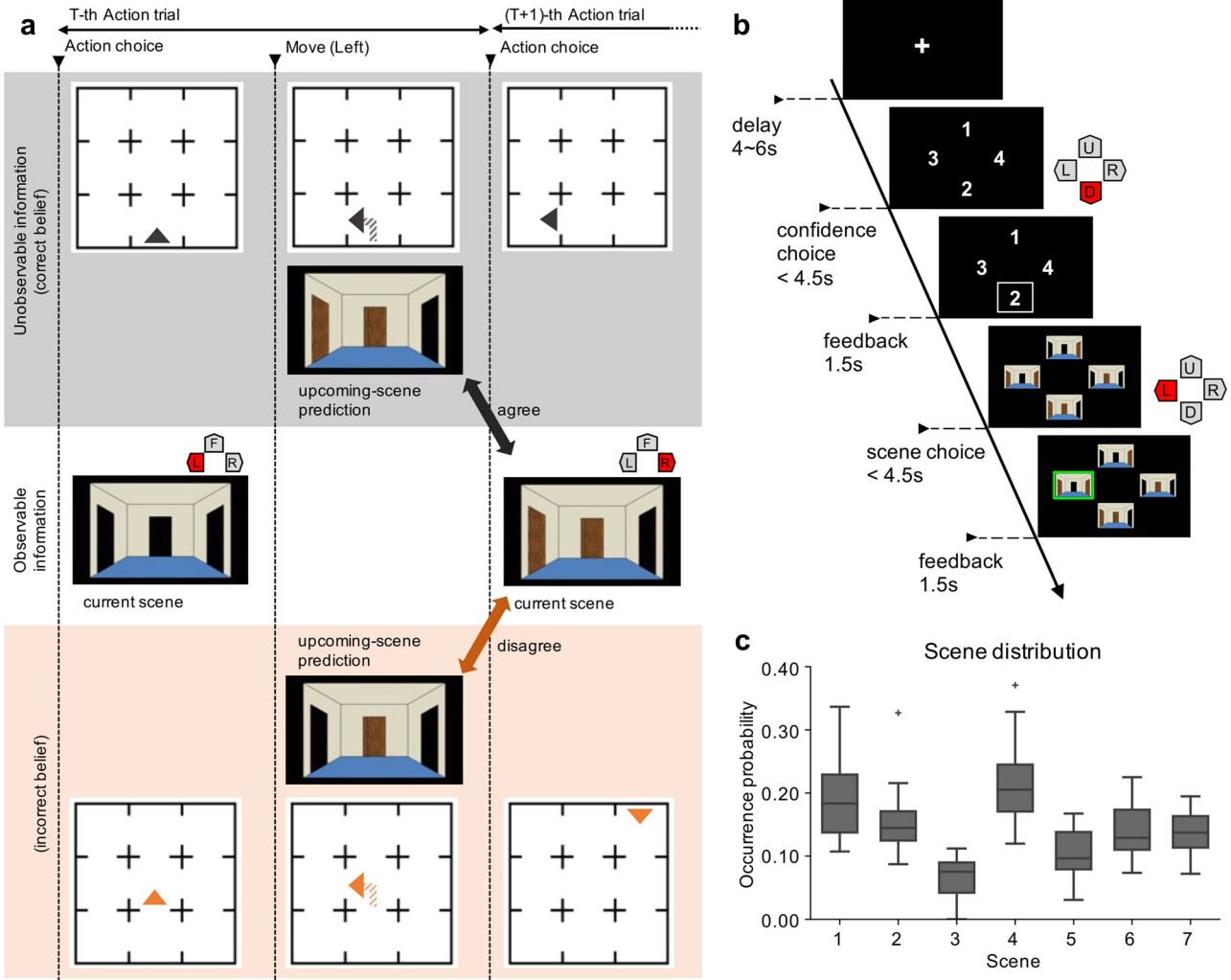

**Fig. 1 Maze navigation task.** Subjects explored the pre-learned grid maze from an unknown initial state and were intermittently asked to predict the upcoming scene and estimate their confidence level for that prediction. To successfully perform the task, subjects needed to infer their current state based on the history of actions and previously observed scenes. **a** A sample action trial sequence. At the beginning of each trial, subjects observed the scene from their current state (i.e., the status of the doors to the left, forward, and right) and then chose an action allowing them to move in one of those three directions. The doors were either open (passable; black) or closed (impassable; brown). Only open doors allowed the subjects to move to an adjacent state in the grid and to see its scene. If a subject's state inference (i.e., belief) was incorrect, the observed scene in the subsequent trial would differ from their prediction. Subjects performed 1–5 consecutive action trials between prediction trials. The 3 × 3 maze in this figure is used to explain the task, and the maze used in the actual experiment was of a 5 × 5 size (Supplementary Fig. S1a). **b** A sample prediction trial sequence. In the prediction trial, a fixation cross was displayed for 4–6 s (delay period) during which time the subjects were asked to predict the upcoming scene. Next, the subjects reported their confidence level for the upcoming scene prediction on a scale from 1 (low) to 4 (high). Then they were asked to select their prediction of the upcoming scene from four options, consisting of the true scene and three distractor scenes. A green or red frame was presented around the selected scene to reflect a correct or incorrect choice, respectively. **c** Occurrence probabilities for each scene selected by subjects as the predicted upcoming scene. There were seven types of scenes based on combinations of door statuses in each scene within the maze (no dead-end, i.e., three closed doors). Center lines of the box plots indicate the medians, boxes indicate the lower and upper quartiles, and the whiskers represent 1.5× interquartile range (IQR). Cross-markers indicate the outliers.

$F(1,5435) = 77.83$, $p = 1.5 \times 10^{-18}$) across all subjects. Similarly, scene choice RTs were shorter in the correct trials than in the incorrect trials ($F(1,5435) = 33.89$, $p = 6.2 \times 10^{-9}$). There was no interaction effect between the prediction correctness and confidence level ($F(1,5435) = 1.8 \times 10^{-2}$, $p = 0.89$).

**Neural correlates of scene prediction.** First, we carried out a univariate general linear model analysis during predicting the upcoming scene (first 4 s of the delay period) and found significantly higher brain (BOLD) responses in the bilateral superior parietal lobules (SPL; Brodmann area [BA] 7), bilateral inferior parietal lobule (IPL; BA40), left dorsal premotor cortex (PMd;

BA6), and left anterior prefrontal cortex (aPFC; BA10) (Fig. 3a; the time-series of the brain activity in each ROI were shown in Supplementary Fig. 3). The statistics are summarized in Supplementary Table 1. One subject was excluded from the imaging and decoding analyses due to his/her larger head motion; accordingly, the following analyses included 26 subjects. We used voxel clusters in the cortical regions above as regions of interest (ROIs) for subsequent decoding analyses.

**Decoding analysis of scene prediction and confidence.** We performed a multi-voxel pattern analysis using the voxel-wise activity patterns of the SPL, IPL, PMd, and aPFC. For the SPL and

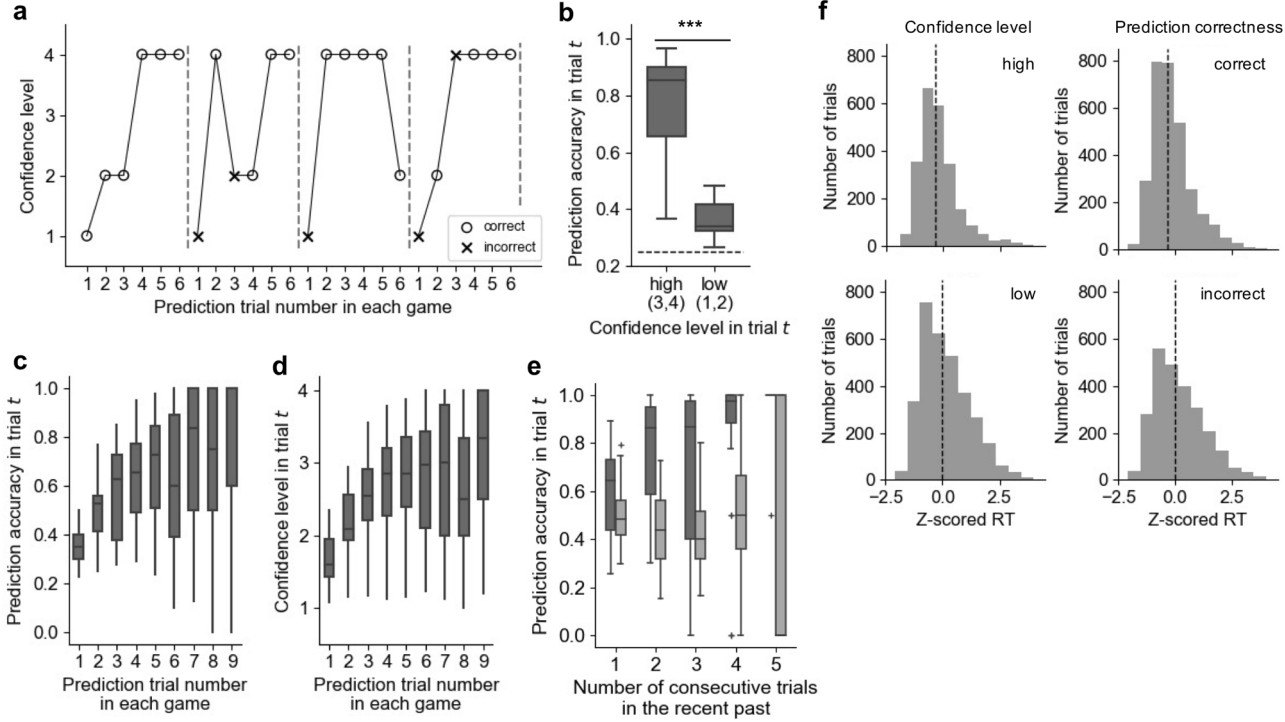

**Fig. 2 Behavioral results. a** Representative example of a sequence of subject's reported confidence levels for scene predictions during four consecutive games. Each marker represents the confidence level chosen in the prediction trial; circle and cross markers denote whether the subject's chosen prediction was correct and incorrect, respectively. The prediction trials are visibly interspersed by action trials. **b** The prediction accuracy was significantly higher when subjects' confidence levels were high (level 3 or 4) on the same trial (one-sided Wilcoxon signed-rank test, ***: $p < 0.001$). The dashed line indicates the chance level. Each box extends from the lower to upper quartiles, with a horizontal line at the median. The whiskers indicate $1.5 \times$ IQR. **c, d** Scene prediction accuracy (**c**) and scene prediction confidence level (**d**) represented as a function of the prediction trial number in each game. Both the prediction accuracy ($r = 0.27$, $p = 1.3 \times 10^{-5}$) and the confidence level ($r = 0.29$, $p = 2.6 \times 10^{-6}$) increased as the number of prediction trials performed in a single game increased. Each box extends from the lower to upper quartiles, with a horizontal line at the median. The whiskers indicate $1.5 \times$ IQR, and cross markers indicate the outliers. **e** Scene prediction accuracy as a function of the number of consecutive correct or incorrect choices in previous prediction trials. The prediction accuracy increased as the number of consecutive correct prediction trials increased (dim gray boxplot, $r = 0.34$, $p = 6.6 \times 10^{-5}$). In contrast, there was no correlation between prediction accuracy and the number of consecutive incorrect prediction trials (light gray boxplot, $r = -0.08$, $p = 0.35$). Each box extends from the lower to upper quartiles, with a horizontal line at the median. The whiskers indicate $1.5 \times$ IQR, and cross markers indicate the outliers. Each data point represents the average accuracy for a single subject. **f** Distributions of the scene choice reaction times (RTs) for all subjects. ART-ANOVA with confidence level (high and low) and prediction correctness (correct and incorrect) revealed that the RTs were shorter in high-confidence trials than in low-confidence trials ($p = 1.5 \times 10^{-18}$) and shorter in the correct trials compared to incorrect trials ($p = 6.2 \times 10^{-9}$). There was no interaction effect between prediction accuracy and confidence level ($p = 0.89$). Each RT was converted to a z-score, normalized within each session for each subject. Vertical dotted lines indicate the median of each distribution. Note that the data shown in **b**–**f** was of 27 subjects included in the behavioral analyses.

IPL, the left and right ROIs were combined, as in the recent decoding studies on navigation[1,2]. As the inputs for both the scene prediction and the confidence decoders, we used brain images of the delay period during which the subjects predicted an upcoming scene in their mind without any visual information. To probe the sensitivity of the decoders to pattern activation time course, we constructed the decoders of the scene prediction and the subject's reported confidence level at nine different time points, the 0th to the 8th decoding periods: the decoders at the $t$-th period used four consecutive scans starting from $t$ s after the onset of the delay period (Supplementary Fig. 4a).

For each decoding period, we constructed six binary classifiers, each of which corresponds to the probability associated with one type of scene, and the scene with the maximum probability was defined as the output of the decoder. In the following decoding analyses, we used the scenes chosen by the subjects for the target labels of the scene decoder regardless of whether they were correct or not. Scene #3 was excluded due to its rarity (Fig. 1c, $13.9 \pm 7.0$ trials, $6.8 \pm 3.2\%$ of the whole). For the confidence decoder, we used a binary classifier with high (confidence level 3

or 4) and low (confidence level 1 or 2) confidence categories. One subject was excluded from the confidence decoding analysis because he/she reported high confidence in three out of 233 prediction trials. Mean decoding accuracies in terms of leave-one-session-out (LOSO) cross-validation (CV) were averaged across all subjects, representing a total of 26 for scene prediction decoding and 25 for confidence decoding.

The MVPA for scene prediction generated classification accuracies that were significantly higher than chance in SPL, IPL, and PMd using a one-sided Wilcoxon signed-rank test (Fig. 3b). However, results from the aPFC were not significant (Fig. 3b, SPL, $18.6 \pm 2.3\%$, $p = 3.3 \times 10^{-4}$; IPL, $18.5 \pm 2.0\%$, $p = 1.9 \times 10^{-4}$; PMd, $17.9 \pm 1.8\%$, $p = 1.1 \times 10^{-3}$; aPFC, $17.2 \pm 1.8\%$, $p = 9.8 \times 10^{-2}$). In contrast, the confidence level could be decoded from activity within all four ROIs (Fig. 3c, SPL, $63.4 \pm 9.2\%$, $p = 1.2 \times 10^{-5}$; IPL, $63.9 \pm 9.0\%$, $p = 1.1 \times 10^{-5}$; PMd, $60.8 \pm 8.1\%$, $p = 2.2 \times 10^{-5}$; aPFC, $59.4 \pm 8.4\%$, $p = 3.4 \times 10^{-5}$). The results of our time-series decoding analysis are shown in Supplementary Fig. 4b–e. There was no significant positive correlation between the number of samples (frequency)

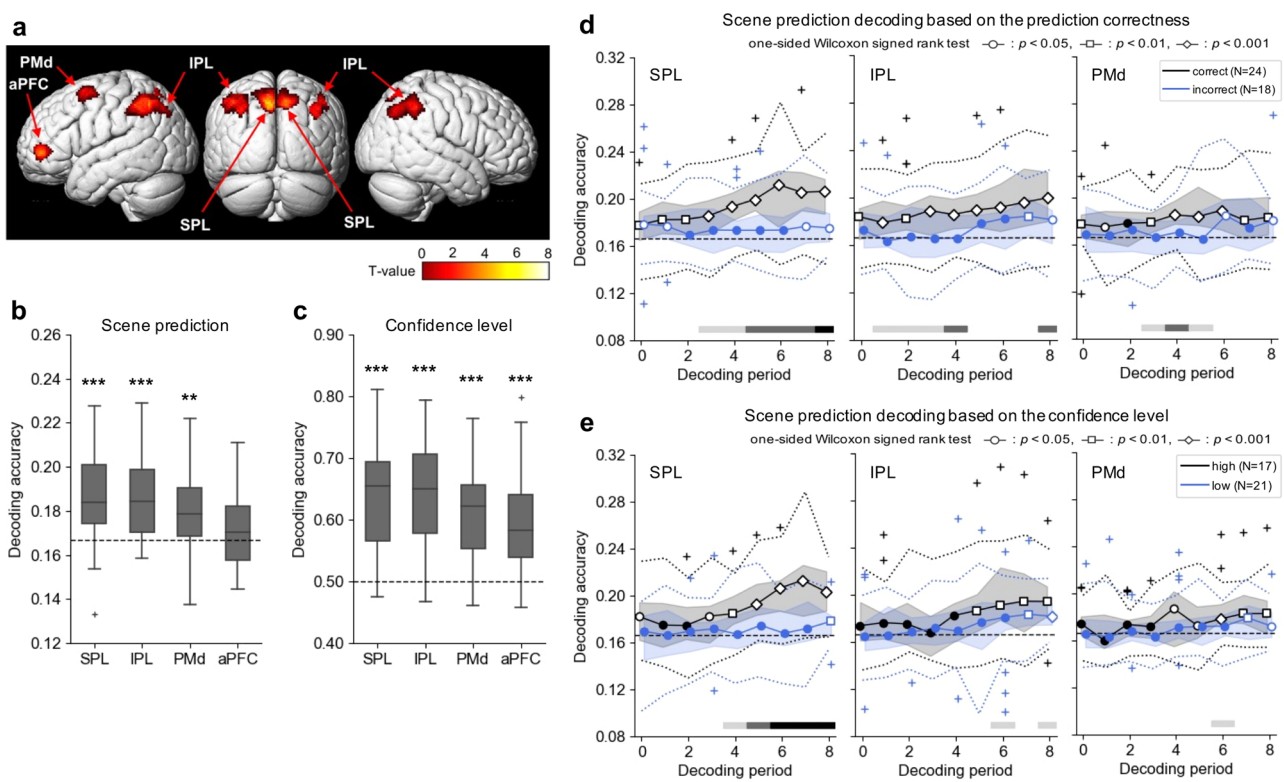

**Fig. 3 Imaging analysis results and decoding accuracies. a** Brain areas that were significantly activated when subjects were predicting an upcoming scene: bilateral superior parietal lobule (SPL), bilateral inferior parietal lobule (IPL), left dorsal premotor cortex (PMd), and left anterior prefrontal cortex (aPFC). The voxel activity patterns in these four areas were used to construct decoders for each region of interest (ROI) for scene prediction and confidence level. Visualization was performed using xjView toolbox (https://www.alivelearn.net/xjview). **b, c** Decoding accuracies for scene prediction (**b**, six types of scenes) and its confidence level (**c**, high or low) within each ROI evaluated using leave-one-session-out (LOSO) cross-validation (CV). Each box extends from the lower to upper quartiles, with a horizontal line at the median. The whiskers represent 1.5 × IQR, and cross markers indicate the outliers. Significance was tested using a one-sided Wilcoxon signed-rank test compared to chance (dashed line) (**: $p < 0.01$, ***: $p < 0.001$). These figures represent the results of the scene prediction and confidence decoders using the 6th decoding period as representative data because of its increasing decodability in our time-series analysis; the overall results of the time-series decoding analysis are shown in Supplementary Fig. 4b–e. **d, e** Time-series scene prediction decoding results within each ROI when the data were categorized binarily according to the prediction correctness (**d**, correct vs. incorrect trials) and the confidence level (**e**, high-confidence vs. low-confidence trials) of the prediction trial. For example, 'correct' indicates the accuracy of the scene prediction decoder trained and tested with only the trials in which subjects' upcoming scene selections were correct (correct-only decoder). The decoding accuracies were evaluated using the leave-one-game-out (LOGO) CV. The solid lines reflect the median, the shaded areas indicate the range between the upper and lower quartiles, and the dotted lines indicate the range of 1.5×IQR. The cross-markers indicate outliers. Significance was tested using a one-sided Wilcoxon signed-rank test (unfilled circle: $p < 0.05$, unfilled square: $p < 0.01$, unfilled diamond: $p < 0.001$) compared to the chance level (dashed line). The color of the horizontal line below the plots reflects a significant difference between the two categories of trials in each decoding period (one-sided Wilcoxon rank-sum test, light gray: $p < 0.05$, dim gray: $p < 0.01$, black: $p < 0.001$).

and the decoding accuracy of each scene in all four ROIs at nine decoding periods (i.e., 36 decoders in total; Supplementary Table 2). Additionally, while the size of ROIs varied from 497 to 2686 voxels, the number of selected features was almost constant between the different ROIs (Supplementary Fig. S4f, g). Therefore, the unbalanced number of samples and dimensionalities of ROIs were found not to distort the decoding results. We also confirmed that there was no positive bias in the decoding accuracy even though the data for decoder evaluation was also used for the ROI selection[21] (Supplementary Fig. S4h, i).

Based on the analysis of scene choice RTs, we expected that the distinctiveness of scene prediction may be different when the subjects had high confidence in the prediction compared to low confidence, and when the subjects successfully predicted the true upcoming scene compared to when they did not. We assessed the accuracy of the scene prediction decoders in the high-confidence versus low-confidence trials and in the correct versus incorrect trials. Here, we constructed four independent decoders trained and tested using different subsets of data: correct or incorrect

prediction trials (regardless of the confidence level), and high- or low-confidence trials (regardless of the correctness) using a leave-one-game-out (LOGO) CV procedure. For each decoder, subjects who had fewer than three samples in the training subset for any individual scene label were excluded from the analysis. Consequently, we used data from 17 subjects for the decoders trained by the high-confidence trials (high-scene-confidence-only decoder), 21 subjects for the low-scene-confidence-only decoder, 24 subjects for the correct-only decoder, and 18 subjects for the incorrect-only decoder. We examined the accuracies in the SPL, IPL, and PMd, which corresponded to the ROIs with acceptable scene prediction decodability.

When comparing the decoding accuracies between the two categories of prediction correctness (correct versus incorrect trials), the three ROIs showed similar patterns of scene prediction decoding accuracies; the decoders tended to exhibit significantly higher accuracies than chance in the correct trials, but not in the incorrect trials (Fig. 3d). However, the voxel activity patterns of the SPL allowed us to decode the subjects' predicted scenes, even

in the incorrect trials, especially in the relatively early stages of the delay period. The differences between the accuracies of the correct-only and the incorrect-only decoders were highly significant for the 5th to the 8th decoding period with the SPL responses, for the 4th and 8th period with the IPL responses, and only for the 4th period with the PMd responses. We also confirmed that the beta estimates and the percent signal changes (PSCs) in the ROIs (SPL, IPL, and PMd) were not significantly different between the compared conditions (Supplementary Fig. 5a, c). Therefore, the differences in the scene prediction decoding accuracy were not due to the activity differences that were examined in the univariate analyses.

When separately decoding scene prediction between the high and low confidence levels, the decodability in the time-series analysis behaved differently between the ROIs (Fig. 3e). When trained with the high-confidence trials, the decoders with the SPL responses exhibited significantly higher accuracy than chance for the 0th and the 3rd to 8th periods, while the accuracy of the low-scene-confidence-only decoder did not differ from chance (except for the 8th period). In addition, for the 6th to 8th periods, the decoding accuracy in the high-scene-confidence trials was significantly higher than that in the low-confidence trials (one-sided Wilcoxon rank-sum test, for the 6th period, $p = 6.8 \times 10^{-5}$; the 7th, $p = 1.5 \times 10^{-4}$; the 8th, $p = 6.9 \times 10^{-4}$). This difference was more pronounced when comparing the trials with the highest confidence level (confidence level 4) with those with the lowest level (confidence level 1) (Supplementary Fig. 5e, for the 3rd period, $p = 2.7 \times 10^{-2}$; for the 4th period, $p = 1.6 \times 10^{-2}$). To assess the relationship between the confidence level and scene prediction decodability, we also compared the decoding accuracy for the 6th period on three scales: the highest (confidence level 4), moderate (2 and 3), and lowest (1). We found that the decoding accuracy significantly increased as the confidence level increased (Supplementary Fig. 5f, $r = 0.52$, $p = 3.8 \times 10^{-4}$). Additionally, we confirmed that there was no significant difference in the beta estimates between the high-scene-confidence trials and the low-scene-confidence trials (Supplementary Fig. 5d). When comparing the PSCs between the high-confidence trials and the low-confidence trials, there was a weakly significant difference from $t = 8$ to $10$ s since the onset of the delay period (Supplementary Fig. 5b). These results indicate that the difference in decodability depending on the confidence level in SPL, especially up to the 7th period, is not due to the difference at the univariate level. On assessing the voxel activity patterns in the IPL and PMd, the scene prediction decoder outperformed chance in the later decoding periods (for the 5th to the 8th period in the IPL; for the 4th to the 8th period in the PMd). However, the difference between the accuracies of the high-scene-confidence-only and the low-scene-confidence-only decoders was weakly significant only in the 6th and 8th periods for IPL (the 6th period, $p = 4.2 \times 10^{-2}$; the 8th, $p = 2.6 \times 10^{-2}$), and only in the 6th period for PMd ($p = 2.8 \times 10^{-2}$). In summary, the correctness of the scene prediction affects the scene prediction decodability with the voxel activity patterns of SPL, IPL, and PMd, while the confidence level of the scene prediction influenced the decodability with the SPL responses only.

**Computational model of maze navigation behavior**. To predict an upcoming scene, subjects must infer their *hidden* current state and then mentally simulate the next state based on their chosen action and the environmental model (maze structure). State inference was inherently uncertain at the beginning of each game, but prediction accuracy was improved as subjects explored the maze more and completed more prediction trials (Fig. 2c). To reproduce the subjects' internal decision-making process, we

implemented a hidden Markov model (HMM) of the subjects' maze exploration behaviors based on previous modeling studies[6,21] (Supplementary Fig. 6). We integrated the following cognitive state variables into our HMM: i) a state inference, which is the belief about one's location and orientation in the maze, and ii) the confidence level for the state inference (i.e., high or low state confidence). We assumed that the subjects used a simple switching mechanism between two strategies depending on their state confidence;[21–23] when they were uncertain about their state inference (low state-confidence level), they moved forward if possible to maximize information to identify the current state (forward-dominant strategy), while when they were certain about their state (high state-confidence level), they tended to move to grid spaces that they had not previously visited (efficient-exploration strategy). In terms of negative log evidence and AIC, this model performed better than the other models with a single strategy (Supplementary Tables 3 and 4).

Figure 4a represents two examples of subjects' behaviors in the maze (left panels) and the most probable paths produced by our model (right panels). The upper and lower panels correspond to subjects with representatively good and poor scene prediction performances, respectively. Our model was good at predicting subjects' actions in the action trials ($95.8 \pm 3.0\%$ overall, and $94.2 \pm 6.6\%$ or $84.7 \pm 12.9\%$ when a scene had two or three open doors, respectively), as well as subjects' scene choices in the prediction trials ($63.3 \pm 15.0\%$). The HMM reproduced the subjects' scene choices in all correct trials (number of correct trials per subject, $112.6 \pm 30.9$), while its reproducibility in incorrect trials was significantly lower than that in correct trials ($21.0 \pm 10.8\%$; number of incorrect trials per subject, $92.7 \pm 32.7$).

The model allowed us to estimate the progression of state-confidence levels for each subject. If the model is reasonable, the state confidence should increase as exploration progresses; if the state-confidence level is high, subjects would be able to make correct scene choices in the prediction trials. The proportion of high-state confidence levels increased as the number of prediction trials increased within a given game (Fig. 4b, $r = 0.63$, $p = 2.6 \times 10^{-28}$), and the prediction accuracy was significantly higher in the high-state-confidence trials than in the low-state-confidence trials (Fig. 4c, one-sided Wilcoxon signed-rank test, $p = 5.8 \times 10^{-6}$).

The state-confidence estimated by our model matched with the scene-prediction-confidence levels reported by the subjects: $63.2 \pm 10.5\%$ for 27 subjects (see also "Discussion"). The state-confidence was also decodable from the voxel activity patterns of four ROIs in which the scene-confidence was found to be decodable (Fig. 4d, one-sided Wilcoxon signed-rank test, SPL, $59.5 \pm 6.0\%$, $p = 7.8 \times 10^{-6}$; IPL, $58.5 \pm 5.9\%$, $p = 5.6 \times 10^{-6}$; PMd, $56.0 \pm 5.3\%$, $p = 2.4 \times 10^{-5}$; aPFC, $55.2 \pm 6.1\%$, $p = 3.3 \times 10^{-4}$). The time-series decoding analyses yielded the same results, regardless of the decoding time points (Supplementary Fig. 7a).

We compared the scene prediction decoding accuracies between the high- and low-state-confidence trials. Two decoders were individually trained using high- and low-state-confidence trials and evaluated using the LOGO CV procedure. Some subjects were excluded from each of the two decoders given the small number of samples ($\leq 2$) for at least one type of scene, resulting in a sample size of 21 subjects for the high-state-confidence-only decoders and 20 subjects for the low-state-confidence-only decoders.

Figure 4e represents the results of the SPL, IPL, and PMd. As an overall trend, the time-series decoding analysis revealed that the scene prediction decoders' accuracy was significantly higher than chance in the high-state-confidence trials within all three ROIs, but not in the low-state-confidence trials, except for the latter decoding period (the 4th to the 8th). The difference in the

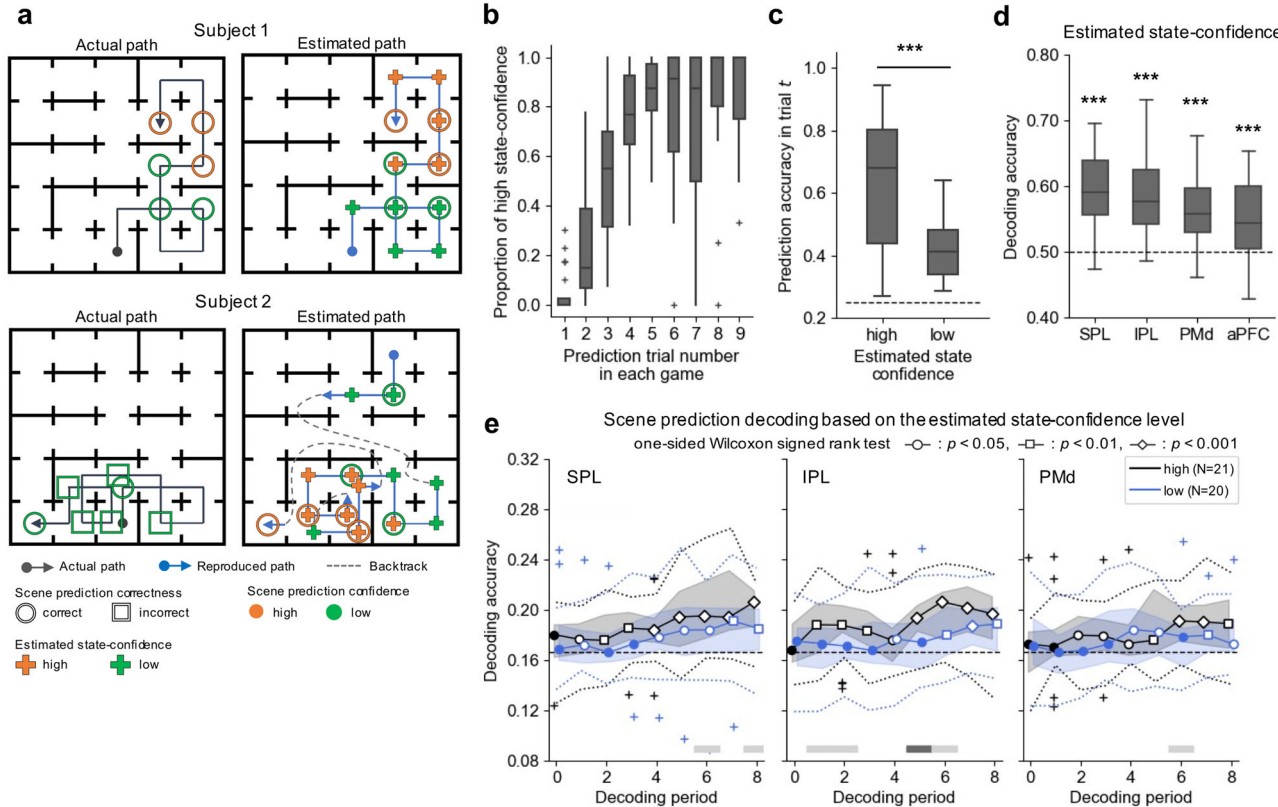

**Fig. 4 Results of behavioral and decoding analyses based on the computational model of human navigation. a** Examples of subjects' actual behaviors in the maze (left panels) and estimated behaviors using a hidden Markov model (HMM) with the maximum a posteriori probability estimate (right panels). Subject 1 and Subject 2 are representative examples of good and poor performances, respectively. The arrows represent the actual (black) and estimated paths (blue), starting from unknown initial positions (filled circles). The dashed lines on the estimated path indicate the subject's position before and after the HMM inferred that the subject had re-estimated their position due to a discrepancy between the expected and observed scenes. The circle or square markers signify the positions in which the subject made a correct or incorrect scene prediction, respectively, and the color corresponds to the subject's reported confidence level about the scene prediction in that position (green: low, orange: high). The color of the cross-markers in the right panels represents the confidence level about the current state estimated by the HMM (green: low, orange: high). **b** Proportion of trials with a high state-confidence level as a function of the prediction trial number in each game. The proportion of the trials in which the HMM estimated the subjects' confidence level as high increased as the number of prediction trials performed in a single game increased ($r = 0.63$, $p = 2.6 \times 10^{-28}$). Each box extends from the lower to upper quartiles with a horizontal line at the median. The whiskers represent $1.5 \times IQR$. The cross-markers indicate the outliers. **c** Scene prediction accuracies compared between trials with high and low state-confidence levels estimated by the HMM. The prediction accuracy was significantly higher when the HMM estimated the confidence level is high compared to low (one-sided Wilcoxon signed-rank test, ***: $p < 0.001$). The dashed line indicates chance level. **d** Decoding accuracy of the HMM estimated state confidence. The decoding accuracy was evaluated using LOSO CV (one-sided Wilcoxon signed-rank test, ***: $p < 0.001$). The dashed line indicates the chance level. Here we plotted the result for the 6th decoding period representatively; the results of our time-series decoding analysis are shown in Supplementary Fig. 7c. **e** Time-series decoding accuracies of scene prediction for different state-confidence levels. Each decoder for each period was trained with the trials of high (or low) state confidence estimated by the HMM, and the accuracy was evaluated by LOGO CV. The solid lines represent the median, shaded areas indicate the range between the upper and lower quartiles, and the dotted lines indicate the range of $1.5 \times IQR$. Cross-markers indicate the outliers. Significance was tested using a one-sided Wilcoxon signed-rank test (unfilled circle: $p < 0.05$, unfilled square: $p < 0.01$, unfilled diamond: $p < 0.001$) compared to the chance level (dashed line). The colors of the horizontal line below the plots reflect the significant differences between the two categories of the trials in each decoding period (one-sided Wilcoxon rank-sum test, light gray: $p < 0.05$, dim gray: $p < 0.01$).

decoding accuracies between the high- and low-state-confidence trials was highly significant when using the voxel activity patterns of IPL for the 5th period (one-sided Wilcoxon rank-sum test, $p = 4.7 \times 10^{-3}$). We also confirmed that the beta estimate in IPL was not different between the high- and the low-state-confidence trials and that the PSCs did not consistently show a significant difference at the time points corresponding to the 5th decoding period (Supplementary Fig. 7a, b). When the decoders were trained with the SPL or PMd responses, there were decoding periods for which the scene prediction decoders exhibited a weaker significant difference in accuracy between the two state-confidence levels.

## Discussion
This study demonstrated that future scenes predicted by human subjects during maze navigation, as well as their corresponding prediction confidence, can be decoded from fMRI activity patterns in localized regions of the prefrontal and parietal cortices. To successfully predict an upcoming scene in the task, subjects needed to infer their current state about which uncertainty was intrinsic due to the partial-observability of the maze. Our decoding target was subjective prediction, which had to be mentally simulated by incorporating subjective inference, given that neither the visual information given by the environment nor the memorized map information was sufficient to independently guide maze exploration behaviors. This highlights the major

contribution of our study: decoding subjective prediction adds to the scope of most previous studies that have focused on decoding aspects of the information provided to the subjects directly, including visual[24,25], tactile[26], and pain[27], or distinctly maintained in the brain, such as memory[28–33], imagination[34–37] and dreaming[38]. Although some recent decoding studies have revealed that predicted visual events are also decodable using fMRI[1,2,5] and electroencephalography[3], they did not take into consideration the prediction uncertainty or confidence. The current study uses ongoing mental simulation in the context of uncertainty and allows us to specifically investigate the neural basis of prediction uncertainty.

Using a decoding analysis for four ROIs that were identified by univariate analysis, we showed that both scene prediction and confidence level were decodable from the voxel activity patterns in the SPL, IPL, and PMd. The time-series analysis also showed that the scene prediction decoding accuracy gradually increased from the 5th to 7th periods, which included the BOLD signals after 4–6 s from the delay onset in the SPL and IPL. Although there was no clear peak or a sharp drop in accuracy following the delay period, this could be due to the wide range of time during which the upcoming scene was predicted, or to the need to maintain the predicted scene throughout the prediction trial to report it in the subsequent scene choice period. Meanwhile, our decoding analysis demonstrated that only confidence could be decoded from the aPFC activity. These results suggest that the aPFC may be involved in encoding subjective confidence, but not the associated prediction. Recent studies have shown that the aPFC is a key brain area involved in the metacognitive assessment. In addition, human fMRI experiments have revealed that this region shows significant activation during self-confidence reporting in perceptual[9] and value-based decision-making tasks[8]. It was also demonstrated that metacognitive accuracy in perceptual decision-making is significantly correlated with the strength of functional connectivity arising from the lateral aPFC[39]. Neuroanatomically, the gray matter volume[40], white matter microstructure[40] and gray matter microstructure[41] in the aPFC are correlated with individual introspective ability. Furthermore, some studies have also reported that perceptual confidence can be decoded from the lateral PFC subregions using MVPA[12,13]. Our results are therefore consistent with those of previous studies, supporting the theory that the aPFC may be an important center for metacognitive processing.

Behaviorally, when a subject's confidence in the scene prediction was high, they tended to make a correct scene choice more quickly than when their confidence was low, which is consistent with previous findings in perceptual judgment tasks, such as motion detection[42,43] and two-choice discrimination tasks[44]. Neurophysiologically, short RTs have been associated with decreased variability of neural activities, such as reproducible activity patterns in rodents[45] and firing rates in nonhuman primates[46]. Furthermore, a human transcranial magnetic stimulation study showed analogous results in which the variability in corticospinal excitability was suppressed in fast-response trials in a bandit task[47]. Based on these findings, we expected that the variability of fMRI voxel activity patterns representing a predicted scene would decrease when the subjects' confidence level was high compared to low, enabling the decoder to distinguish the voxel patterns more accurately.

When we assessed the effects of confidence on the decodability of scene prediction, we found that the scene prediction decoders trained with the SPL activities in the high-scene-confidence trials performed significantly better than those with the low-scene-confidence trials, especially in the 6th to the 8th decoding period. An additional decoding analysis was performed by excluding one multiplicative condition ([correct, incorrect]×[high, low]) out of

$2 \times 2$, and it was suggested that the neural representation of scene prediction in SPL was affected by both the confidence level (Supplementary Fig. 5g, ART-ANOVA, 6th period, $F(1,96) = 9.37$, $p = 2.9 \times 10^{-3}$; 7th period, $F(1,96) = 4.25$, $p = 4.2 \times 10^{-2}$) and the prediction correctness in the 6th and 7th decoding periods (6th period, $F(1,96) = 4.10$, $p = 4.6 \times 10^{-2}$; 7th period, $F(1,96) = 4.42$, $p = 3.8 \times 10^{-2}$). Previous studies have demonstrated that MVPA decoding accuracy is influenced by some behavioral and perceptual performance measures, such as tactile discrimination ability[26], spatial memory accuracy[48], reaction time[49], and familiarity with auditory stimuli, both intra-[50] and inter-individually[26,48,49]. Moreover, MVPA has shown that the neural distance, which is the distance from a classifier hyperplane in the activation space, is robustly correlated with RT in the context of categorization tasks[51,52]. Thus, this study adds to the existing literature by uncovering a clear relationship between decoding accuracy and subjective reporting of metacognition.

The prediction confidence level affected the scene prediction decoding accuracy in the SPL, whereas there was no effect on the decodability in the IPL. These results suggest that SPL represents the predicted scene itself and that IPL could be involved in other processes related to scene prediction. Based on the *state-confidence* level estimated by our HMM, the IPL scene prediction decoder was found to perform better in high-state-confidence trials, while the SPL decoder did not differ between high and low state-confidence levels (Fig. 4e). We also determined that only the IPL scene prediction decoder, excluding the correct and high-state-confidence trials, failed to show higher accuracy than chance (Supplementary Fig. 7d). These results support that state confidence partly affects the decoding accuracy of scene prediction with IPL responses. In addition, when we looked at the time-series of decodability, the IPL scene prediction decodability in the high-state-confidence trials reached a peak slightly earlier (the 6th period) than the SPL scene prediction decodability in the high-scene-confidence trials (the 7th period). These results suggest that IPL may be related to the neural representation of state inference, which functions as upstream information processing when predicting the upcoming scene.

The decoding accuracies of scene prediction in the IPL and SPL were thus influenced by the confidence level for state inference and scene prediction, respectively. The parietal cortex is known to be involved in spatial information processing, including navigation[16–19] and there is abundant evidence from human fMRI[53–58], nonhuman primate[59–62], and rodent physiological studies[63] that the posterior parietal cortex is crucial in egocentric forms of spatial information processing. In contrast, IPL has been shown to be associated with allocentric spatial information processing, such as viewer-independent spatial memory[64], object-based spatial judgement[65,66], and object-based attention[67]. Positron emission tomography studies have also shown that the IPL may be involved in the conversion of allocentric-to-egocentric spatial representation in navigation tasks[68]. Our decoding results could be interpreted to suggest that IPL represents the allocentric state inference based on a memorized map, possibly encoded in the hippocampus, while the predicted scene is encoded in SPL as egocentric spatial information for navigation. Although this is speculative and our current data are not suitable for further detailed analysis, the decoding of subjects' state inference, that is, subjects' belief of the position in the maze, may possibly support our interpretation.

In contrast to the SPL and IPL, the PMd decoding accuracy of scene prediction was affected by neither scene-prediction-confidence nor state confidence, although it was significantly higher than chance in the correct trials. Previous studies have reported that the PMd showed significant activation related to action selection[69] and significant decodability of movement-related information such as

hand[70] and target positions[71]. Based on these studies, we speculated that PMd might specifically be involved in action planning in our maze exploration task because a scene is defined by a set of open (passable) or closed (impassable) doors in our experimental design. Unfortunately, we found no further support for this speculation in this study.

Our HMM-based behavioral model allowed us to successfully estimate subjects' state confidence, although it was not explicitly reported by the subjects in the experiment. According to our view of an incremental Bayesian filtering process, scene prediction should incorporate a mental simulation process utilizing the memory of the maze structure after inferring the current state in the maze. Since the scene prediction follows the state inference, it includes additional uncertainties that vary from subject to subject, such as the degree of the imperfectness of the maze structure memory. The subject-reported scene-prediction-confidence may not necessarily match the estimated (but objectively reconstructed by our model) state confidence, as represented in the subjects' brain. In fact, the agreement between the scene-prediction confidence (subject-reported, high or low) and the state confidence (model-estimated, high or low) was $63.2 \pm 10.5\%$ for all prediction trials, suggesting that there may be a difference between these two types of confidence. The difference in the time-course decodability between the SPL and IPL corresponded to this difference between the confidence types.

This study demonstrated that the localized neural representations of predictions during maze exploration differed depending on the confidence level of prediction. In addition, we noted that SPL and IPL appear to have different involvements in scene prediction. However, there are some limitations that should be addressed. Considering the assumed decision-making process underlying maze exploration, it is plausible that confidence or uncertainty about scene prediction could be affected by prior knowledge (i.e., the memory of the maze structure), which may not be perfect. Accordingly, future studies should probe and verify the effects of imperfect prior knowledge on confidence. Moreover, the detailed process of generating confidence from state inference and scene prediction has yet to be explored. It would be an interesting avenue for future studies to consider and verify how the parietal-prefrontal downstream circuit, which includes the IPL, SPL, and aPFC that would hierarchically encode prediction during maze exploration, decodes uncertainty and subjective confidence, to better enhance our understanding of the neural substrates involved in the decision-making process in uncertain environments.

## Methods

**Subjects**. Thirty-three healthy subjects were recruited to participate in the experiment and provided written informed consent. This study was approved by the ethical committees of the Advanced Telecommunications Research Institute International, Japan, and the Graduate School of Informatics, Kyoto University, Japan. Six subjects whose scene prediction accuracy in the experimental task was not significantly higher than chance (one-sided $z$-test, $p \geq 0.01$, see also Supplementary Fig. 2) were excluded from analyses. Another subject was also excluded from imaging and decoding analyses due to his/her large head motion (more than 5% of TRs at Framewise Displacement threshold 0.5 mm). No statistical method was used to predetermine the sample size, but the sample size for our analyses was comparable to those generally employed in the field.

**Maze exploratory navigation task**. The experiment consisted of two tasks: a training task outside the fMRI scanner to learn the structure of the maze (see "Training task"), followed by a maze scene prediction task inside the fMRI scanner. We used a single $5 \times 5$ grid maze for all subjects, in which each grid had either an open (passable) or closed (impassable) door on all four sides (Supplementary Fig. 1)[72]. We used the same maze for all subjects so that there would be no difference in the level of task difficulty between individuals. The maze was partially observable such that at each state (position and orientation), subjects could only observe the current scene (i.e., the status of the doors to the left, right, and forward)[2,6,21]. We created a maze to satisfy partial observability, that is, at least two consecutive observations from the initial state are required to identify the current state. The experimental and training tasks were programmed using Psychopy3[73].

In the scene prediction task, subjects freely explored the maze and were intermittently asked to predict the upcoming scene and to rate their level of confidence about the prediction (Fig. 1). Each game started in an unknown initial state (i.e., a combination of position and orientation), with the current 3D scene (i.e., the status of the doors to the left, forward, and right) displayed on a screen. Subjects were requested to choose an action to move to the left, forward, or right grid space by pressing a button within 2.5 s, and subsequently, the 3D scene at the next state was presented in the next trial (Fig. 1a). If subjects chose an impassable door, they remained in the same state and the same scene was presented again in the subsequent trial ($2.7 \pm 3.1$ trials for 27 subjects, $0.6 \pm 0.7\%$). If an action was not taken within the allotted time, one of the passable doors was chosen by the computer ($12.7 \pm 9.8$ trials, $2.8 \pm 2.0\%$). After repeating an *action trial* 1–5 times, a *prediction trial* was performed (Fig. 1b). In a prediction trial, a fixation cross was displayed for 4–6 s (delay period) instead of the next scene presentation, and subjects were requested to predict the upcoming scene. In principle, although the upcoming scene was determined by the state and action in the previous trial, the intrinsic uncertainty of each state in the partial-observable maze required subjects to make inferences about their current state based on the history of actions and observed scenes. After the delay period, subjects were first asked to evaluate their level of confidence about their upcoming prediction on a four-point scale (1: lowest confidence; 4: highest confidence). The display positions of the four options were randomized for each prediction trial, and a white frame appeared around the selected option as feedback (1.5 s). After the confidence report, four out of seven possible scenes were displayed, and the subjects were requested to select the scene corresponding to their prediction of the upcoming scene. The scene options always included the correct upcoming scene and three other scenes, and their display positions on the screen were randomized. Distracters were selected from the set of possible scenes in which the local structure was consistent with the true state (i.e., states that could be reached from scenes similar to the scene presented in the previous trial using the selected action). Here, the design included four out of seven existing scenes as the predicted scene options, rather than the complete set of existing scenes, to make it easier for subjects to report their prediction by choosing one option within the time limit. Importantly, subjects were asked to report their confidence level prior to the presentation of the scene options, because there is a possibility that the limited options induce a large prediction in their prediction confidence. After selecting a scene option, a green or red frame appeared around the chosen scene, indicating that the choice was correct or incorrect, respectively. In the next action trial, the scene of the true subsequent state was presented, irrespective of the correctness of the subject in scene prediction. The allotted time for both reporting confidence level and selecting an upcoming scene option was limited to 4.5 s in order for subjects to make each choice as soon as possible.

Once a subject reached the termination condition, a yellow star was displayed on the door, leading to the final state. A termination condition was reached if the subjects were experiencing the state for the first time and had performed at least five prediction trials. Subjects were not provided details regarding the termination condition, but they were informed that both the initial and final states varied between games. Each game consisted of 5–20 blocks ($5.5 \pm 1.2$), with each block consisting of 1–5 action trials and one prediction trial. The number of action trials was randomized for each block. Each subject performed up to 40 total games ($38.2 \pm 4.2$ games, $208.5 \pm 19.5$ prediction trials), which were divided into three or four sessions.

In our experimental design, the subjects judged their decision confidence before indicating their predicted scene. One may note the effect of this *prospective* confidence evaluation, since, in many previous studies, confidence was reported retrospectively. A previous study suggested that prospective confidence is more discrepant from objective performance than retrospective confidence[74], but another study reported that there was no significant difference in the confidence rating when decision making was not communicated but performed before confidence judgement[75]. In our current study, subjects performed the confidence report followed by the predicted scene choice to prevent their scene prediction and confidence judgment from being modified after the presented scene options.

**Training task**. Subjects performed multiple training sessions to sufficiently learn the structure of the maze outside of the fMRI scanner on the day before or the same day as the scanning experiment. If the subjects performed the training sessions on the previous day of the scanning experiment, they received re-training on a short version of the training task (one or two training sessions) to avoid the confounding effects of forgetting. One training session consisted of five games in two parts: the first three games were in the practice part in which subjects explored the maze while referring to a printed 2D map, and the latter two were in the test part where they explored the maze without the map and were occasionally asked to predict the upcoming scene.

In the practice games, a printed 2D map of the maze was given to each subject, and they were free to refer to it at any time. At the beginning of each game, the 2D maze map showing the initial state was displayed on a computer screen for 5 s; the initial state, comprised of both position and orientation, was depicted as a red arrow on one of the grid squares. The initial state varied between games. Subjects were then given unlimited time to select a movement action from the initial state

by pressing a button, after which the 3D scene corresponding to the next state was presented. When the subjects reached a termination condition, a yellow star appeared on a door leading to the final grid. A game was terminated if subjects visited a grid for the first time in the game after they performed twenty action selections. Subjects were not instructed about the termination condition but were informed that the final state varied between games. In the practice part, subjects thus performed only the action trials in the maze navigation task.

The test games were similar to the maze navigation task but without confidence ratings. Here, subjects performed the scene prediction task from an unknown initial state and were not able to refer to the printed map. Each trial began with the 3D scene corresponding to the current state displayed on the screen and subjects were given 2.5 s to select an action. After several action trials, subjects were required to perform a prediction: a fixation point was displayed for 4 s (delay period) and then were given 4.5 s to choose the option corresponding to their prediction of the upcoming scene from four options. Like the experimental task, the options consisted of the true upcoming scene and three distractor scenes randomly selected from the set of scenes with a similar local structure to the correct option, and then a green (correct) or red (incorrect) frame was presented around the selected scene as the choice feedback. The subsequent action trial displayed the true upcoming scene, irrespective of the subject's scene prediction correctness. We refer to a set of 1–5 action trials followed by a prediction trial as a block, and subjects repeated these blocks until they reached a termination condition. The first block included ten action trials, allowing the subjects to explore the maze to gather more information about their position in the maze before the first prediction trial. The termination conditions were identical to those used in the scanning experiment. The subjects performed two games in the test part, each of which consisted of 5–48 blocks ($7.0 \pm 4.2$).

All subjects performed at least seven training task sessions, up to a maximum of 1.5 h in total. If a subject's scene prediction accuracy averaged across two test games exceeded 80%, he/she was allowed to end the training task. The best prediction accuracy was $79.2 \pm 19.9\%$ for all 33 subjects, $85.3 \pm 16.2\%$ for 27 subjects who were included in the behavioral analysis, and $85.0 \pm 16.5\%$ for 26 subjects who were included in the imaging and decoding analyses. Here, the best prediction accuracy means the best accuracy for each subject across all sessions, while session accuracy was averaged over two test games.

**Image acquisition and analysis.** A 3.0-Tesla Siemens MAGNETOM Prisma fit scanner (Siemens Healthineers, Erlangen, Germany) with a standard 64 channel phased array head coil was used for image acquisition. We acquired interleaved T2*-weighted echo-planar images (EPIs) (TR, 1000 ms; TE, 30 ms; flip angle, 50°; matrix size, $100 \times 100$; field of view, $200 \times 200$; voxel size, $2 \times 2 \times 2.5$ mm; number of slices, 66). Volume acquisition was synchronized with the onset of the fixation cross-presentation during each prediction trial. We also acquired whole-brain high-resolution T1-weighted structural images using a standard MPRAGE sequence (TR, 2250 ms; TE, 3.06 ms; flip angle, 9°; field of view, $256 \times 256$; voxel size, $1 \times 1 \times 1$ mm).

Imaging data were analyzed using SPM12 (Wellcome Department of Cognitive Neurology, London, UK). For each subject, all functional images were aligned to the first image as a reference, coregistered to the individual high-resolution anatomical image, normalized into an MNI template, and spatially smoothed with a Gaussian kernel filter (FWHM, 8 mm).

Our univariate analysis was based on the generalized linear model (GLM) approach. Our GLM included seven regressors coding for onsets and durations of events in each session: action selection and moving scenery in the action trials, delay period, confidence evaluation, feedback for confidence evaluation, predicted scene choice, and scene choice feedback in the prediction trials. The durations of choice-related events (action selection, confidence evaluation, and predicted scene choice) were defined as the time between the option presentation and the subject's response. The durations of moving and choice feedback were fixed to 1.5 s. For the delay period (regressor-of-interest), although the time length varied trial-by-trial, we modeled it as a boxcar function for 4 s (the minimum duration of the delay period). These regressors were convolved with a hemodynamic response function (HRF). Additionally, motion correction parameters produced during realignment were included as nuisance variables for the GLM. The first-level GLM analysis was performed using the contrast vector whose element was 1 for the regressor-of-interests, and 0 otherwise. We then performed a group random effect analysis using anatomically localized cerebral cortex to find cortical voxels that were significantly and commonly activated during prediction across all subjects. We established statistical thresholds at the voxel level of $p < 0.001$ (uncorrected) and at the cluster level of $p < 0.05$ (FWE-corrected). We extracted regions of interest (ROIs) from the identified voxels, and the BOLD signal patterns in each ROI were used for the decoding analysis.

To complement the univariate ROI analysis, we also performed whole-brain searchlight analyses (see "Searchlight analysis for the scene prediction and confidence"). For scene prediction, we confirmed that there were no clusters other than the four ROIs extracted by the univariate analysis that showed significantly higher decoding accuracy than chance (voxel level, $p < 0.001$; cluster level, FWE-corrected $p < 0.05$). For confidence level, we found that the regions where the confidence can be decoded were widely distributed over the cerebrum, which may

be attributed to the task design in which subjects were instructed to predict the upcoming scene during the delay period.

**Decoding analysis.** Voxel activity patterns during the delay period were used to decode both scene prediction and confidence. All fMRI data were spatially realigned, normalized, and smoothed with a Gaussian kernel (8 mm FWHM), and preprocessed with linear trend removal and z-score normalization for each voxel in every run over the time series but not convolved with HRF.

In the time-series decoding analysis, the decoder at each time $t$ in the time course ($t$-th decoding period) used as its input the voxel-wise BOLD signal intensities averaged over four volumes corresponding to $t$ s to $t + 3$ s (i.e., $(t + 1)$-th to $(t + 4)$-th scan volumes) after the onset of the delay period (Supplementary Fig. 4a). We limited the time-series decoding analysis up to the 8th point in order to cover 4–6 s after the delay onset, which exhibited the peak brain activity evoked by the delay cue, and to ensure that the scene prediction decoder was as unaffected as possible by information provided in the scene choice period (average 7.5 s after the delay onset), during which the subjects' predicted (chosen) scene was displayed.

For the scene prediction decoder, there were seven possible scenes. Each scene was labeled with a 3-bit binary number in which each bit corresponded to the status (open: 1; closed: 0) of the left, forward, and right door, respectively. We used the scenes chosen by the subjects for the target labels of the scene decoder regardless of whether they were correct or not. Label 3 (011 in binary code) was excluded from the analysis because of its rare occurrence (Fig. 1c); thus, we used six labels for the decoding analysis. The label for the confidence decoder was either low (confidence level 1 or 2) or high (confidence level 3 or 4). One subject was excluded from the confidence decoding analysis because he/she reported high confidence in only three trials (1.3%).

We used a sparse logistic regression (SLR)[76] as a supervised learning algorithm because it incorporates Bayesian automatic selection of relevant features (voxels), which prevents overfitting problems in high-dimensional neuroimaging data. This method has been used for MVPA in the previous studies[2,12,24]. Scene prediction decoders included six scene labels, and we used six one-versus-the-rest classifiers with SLR as the decoder, where a classifier for scene $k$ outputs the probability that the input brain activity pattern $x$ represents scene $k$, $P(\text{scene} = k; x)$, and scene $k'$ with the maximum probability among six classifiers, $k' = \text{argmax } P(\text{scene} = k; x)$, is defined as the integrated decoder output.

To deal with unbalanced training data sets (Fig. 1c), we used an undersampling method to assign an equal number of samples to each label. Although the trial numbers were actually unbalanced between the compared conditions when training the classifiers in the conditional decoding analyses, there were no significant differences in the number of samples: between correct ($51.3 \pm 18.1$ trials) and incorrect trials ($47.7 \pm 17.9$ trials, one-sided Wilcoxon rank-sum test, $p = 0.29$), between high-confidence ($45.9 \pm 17.9$ trials) and low-confidence trials ($61.0 \pm 32.6$ trials, $p = 6.7 \times 10^{-2}$), between high-state-confidence ($55.9 \pm 25.8$ trials) and low-state-confidence trials ($53.4 \pm 15.8$ trials, $p = 0.51$).

To assess decoder accuracies, we used LOSO CV in which each decoder was trained using a training data set from three out of four sessions, and the remaining session was used as test data for validation. When evaluating the decoders with the trials divided into two categories according to confidence or correctness, we used LOGO CV. In each fold of the LOSO and LOGO validations, we repeated the following procedure 100 times to account for fluctuations in accuracy due to selected samples in the undersampling phase: random under-sampling from the training data set, training the decoder, and evaluating the decoder's accuracy. The decoding analysis method was implemented using Brain Decoder Toolbox[77].

**Searchlight analysis for the scene prediction and confidence.** To complement the univariate ROI analysis, we conducted whole-brain searchlight analyses with 10 mm radius spheres centered around a given voxel for the session-wise unsmoothed beta estimates. In the scene prediction searchlight analysis, the seven different scenes predicted by the subjects were modeled in the GLM as seven individual regressors during the delay period. Other than the delay period, we used the same regressors as in the original GLM. For the confidence level, we used the GLM with two regressors according to the subject's confidence level (high or low) in the delay period. To create a subject-level whole-brain accuracy map, we used a linear support vector machine[78] and the accuracy of each voxel was evaluated using leave-one-session-out (LOSO) cross-validation (CV). The individual accuracy maps were normalized and smoothed using a Gaussian kernel (8 mm FWHM)[11,79] and then applied to the group random effect analysis using anatomically localized cerebral cortex.

**Permutation test for the scene prediction and confidence decoding analyses.** We also performed a two-step permutation test to test the null hypothesis that the decoding accuracy of scene prediction and confidence was not different from the chance level, and confirmed that the decoding accuracies in our analyses do not exceed the theoretical chance level merely by chance[80]. Within each ROI for each decoding period, first (i) we performed intra-subject permutation, i.e., repeated each decoding analysis $N_{sbj}$ times in each of which labels to be the decoding targets were randomly permuted within sessions for each subject. We used $N_{sbj} = 150$. Then (ii) we drew for each subject one result randomly from the pool of these permutation results including the original decoding accuracy, and

averaged among the subjects to calculate the group-level permutation result. (iii) Step (ii) was repeated $N_{group}$ times to acquire the null distribution for group statistical analysis. We used $N_{group} = 1000$. The significance of the difference between the group-level null distribution and the original decoding accuracies was tested using a one-sided Wilcoxon rank-sum test. We did not perform the permutation tests to inspect the significance of the conditional decoding analysis results because they were found to require too long a computation time. Note, however, that our under-sampling technique has made the prior of labels for each of the conditional decoders uniform. We also assessed the significance of the decoding accuracies of scene prediction and confidence through group-level permutation testing and confirmed that our decoding results were not disturbingly high (Supplementary Fig. 4d, e).

**Behavioral model based on a HMM.** We constructed a behavioral model for each subject based on an HMM with a latent variable denoting confidence level. The model simulates a generative process of a subject's actions ($a^*$) based on the sequence of observed 3D scenes ($o^*$) and internal cognitive states. A subject's cognitive state was modeled as the state inference (location and orientation) within the maze ($h$), the operant state ($m$), and the confidence level about the state inference (or state confidence, $c_h$). In action trial $t$ during the maze navigation task, subjects were assumed to probabilistically alter their state confidence ($c_{h,t}$), subjectively infer the hidden true state ($h_t^*$) as its estimate from the history of observations ($h_t$), and then select an action ($a_t^*$) based on a decision strategy determined by the state confidence. To efficiently reach an unspecified goal (i.e., with as few actions as possible), it is essential to estimate the hidden true state and visit previously unexplored areas of the maze. We hypothesized that subjects switched between two action selection strategies depending on the cognitive state, based on a previous study:[21–23] one is a forward-dominant strategy and the other is an efficient-exploration strategy. Throughout the behavioral model, a variable with an asterisk (*) is real (physical), observable, and objective, and a variable without an asterisk is internal (cognitive), unobservable, and subjective. A variable with a hat (^) explicitly denotes a prediction.

We developed the subject behavioral model as follows (Supplementary Fig. 6): at $t = 1$, all possible states that are consistent with the first scene ($o_1^*$) were extracted as candidates of the hidden true state, where $H_1$ is the set of those states. One of the sets was chosen as an initial state estimate ($h_1$), for which the posterior probability was expressed as $P(h_1) = 1/|H_1|$. Here, $|H_1|$ is the number of elements in $H_1$. At the start of a game, subjects were assumed to have low confidence level about their state inference, so the state confidence was set at 0 with a probability one: $P(c_{h,1} = 0) = 1$ (Step 1). At $t \geq 1$, the action selection strategy ($\pi_t$) was determined using the current state confidence ($c_{h,t}$). If the state confidence was low ($c_{h,t} = 0$), $\pi_t$ was set as the forward-dominant strategy; if the state confidence was high ($c_{h,t} = 1$), $\pi_t$ was set as the efficient-exploration strategy (Step 2). According to $\pi_t$, an action ($a_t^*$) was probabilistically selected based on $h_t$. The action strategy is described in further detail below (Step 3). A new state estimate was calculated as $\hat{h}_{t+1}$, based on $h_t$ and $a_t^*$, accounting for the maze structure. Then, the upcoming scene was predicted as $\hat{o}_{t+1}$, based on $\hat{h}_{t+1}$ (Step 4). After the subjects moved to the true next state ($h_{t+1}^*$)— based on the previous state ($h_t^*$) and action ($a_t^*$)—the true next scene ($o_{t+1}^*$) was observed. Note that the real state ($h_{t+1}^*$) may differ from its estimate ($\hat{h}_{t+1}$) (Step 5). If $\hat{o}_{t+1}$ matched $o_{t+1}^*$, the subjects were assumed to consider their previous inference to be confirmed. This was called an update mode, represented by $m_{t+1} = 0$. In this case, $\hat{h}_{t+1}$ was subjectively confirmed as the new state inference, $h_{t+1} = \hat{h}_{t+1}$. Concurrently, the state confidence $c_{h,t}$ was updated to $c_{h,t+1}$ stochastically with the transition probability ($P_{UD}$). The transition of the state confidence is described in further detail below (Step 5a). If $\hat{o}_{t+1}$ did not match $o_{t+1}^*$, the subjects were assumed to dispose their previous inference $\hat{h}_{t+1}$. This was called a backtrack mode, represented by $m_{t+1} = 1$. In this case, a new set of states ($H_{t+1}$) was constructed to account for the current observation ($o_{t+1}^*$) and the history of past observations. A new state estimate was randomly chosen from $H_{t+1}$ as $h_{t+1}$, for which the posterior probability was expressed as $P(h_{t+1}) = 1/|H_{t+1}|$. The new state confidence ($c_{h,t+1}$) was stochastically determined, depending on $c_{h,t}$ and according to the transition probability ($P_{BT}$) (Step 5b). The procedure then backs to step 2 with $t \leftarrow t+1$.

Subjects were considered to have low state confidence ($c_{h,t} = 0$) when they were uncertain about their state in the maze (Supplementary Fig. 6, Step 3). In this case, the subjects were assumed to take "info-max" behaviors to efficiently identify the state; in other words, they moved forward or chose an action at random if a forward move was not possible. This *forward-dominant strategy* was defined as follows: if the door in front of the subject was open, the forward movement was considered the optimal action. If the door in front of the subject was closed and both the left and right doors were open, the right move was considered the optimal action (based on retrospective reports from all subjects). If there was only one open door, the action in the passable direction was considered optimal.

We assumed that the subjects' action selection was probabilistic and that they chose an optimal action with probability $\alpha$ as follows:

$$P(a_t|h_t, c_{h,t} = 0) = \begin{cases} \alpha & \text{if } a_t \text{ is optimal} \\ (1-\alpha)/N_{nopt} & \text{otherwise} \end{cases} \quad (1)$$

$N_{nopt}$ denotes the number of allowable (i.e., passable doors) but non-optimal actions.

On the other hand, when the state confidence was high ($c_{h,t} = 1$), subjects were considered to be certain about their state in the maze (Supplementary Fig. 6, Step 3). In this case, subjects preferentially moved to grid spaces that they had not yet explored. This *efficient-exploration strategy* was defined as follows: if there was only one open door, the optimal action was in the single passable direction. If there were two or three open doors, the subjects preferentially chose the door leading to an unexplored grid space: if there were one or more accessible adjacent grid spaces that the subjects had yet to visit, all actions leading to the unexplored grids were considered optimal. If all of the accessible grid spaces had been visited, the optimal action was considered to be choosing the shortest path to the nearest unexplored grid space.

The action selection probability in the efficient-exploration strategy is defined as follows:

$$P(a_t|h_t, c_{h,t} = 1) = \begin{cases} \beta/N_{opt} & \text{if } a_t \text{ is optimal} \\ (1-\beta)/N_{nopt} & \text{otherwise} \end{cases} \quad (2)$$

where $\beta$ is the probability of optimal action selection, and $N_{opt}$ and $N_{nopt}$ denote the numbers of optimal actions and allowable but non-optimal actions, respectively.

We assumed that the subjects stochastically changed their state confidence levels (high or low) based on whether or not their scene prediction matched the observed scene. If the observation ($o_{t+1}^*$) agreed with the scene prediction ($\hat{o}_{t+1}$), the subjects were assumed to become more confident about their state estimate, whereas if $o_{t+1}^*$ disagreed with $\hat{o}_{t+1}$, they were assumed to become less confident.

More concretely, when the subjects had low confidence levels about their state estimate ($c_{h,t} = 0$) but there was no discrepancy between the predicted scene and the observed scene (update mode; Supplementary Fig. 6, Step 5a), their confidence level was switched to high with a probability of $p_{L\rightarrow H}$. If the state confidence was already high ($c_{h,t} = 1$), it stayed high. In contrast, when the subjects were confident about their state estimate ($c_{h,t} = 1$) but the predicted scene differed from the observed scene (backtrack mode; Supplementary Fig. 6, Step 5b), the confidence level was switched to low with a probability of $p_{H\rightarrow L}$. If the state confidence was already low ($c_{h,t} = 0$), it stayed low.

In summary, the dynamics of the confidence level were defined by a Markov process depending on the operant state:

In the update mode ($m_{t+1} = 0$),

$$\begin{pmatrix} P(c_{h,t+1} = 0) & P(c_{h,t+1} = 1) \end{pmatrix} = \begin{pmatrix} P(c_{h,t} = 0) & P(c_{h,t} = 1) \end{pmatrix} P_{UD}$$
$$\text{where} \quad P_{UD} = \begin{pmatrix} 1 - p_{L\rightarrow H} & p_{L\rightarrow H} \\ 0 & 1 \end{pmatrix} \quad (3)$$

In the backtrack mode ($m_{t+1} = 1$),

$$\begin{pmatrix} P(c_{h,t+1} = 0) & P(c_{h,t+1} = 1) \end{pmatrix} = \begin{pmatrix} P(c_{h,t} = 0) & P(c_{h,t} = 1) \end{pmatrix} P_{BT}$$
$$\text{where} \quad P_{BT} = \begin{pmatrix} 1 & 0 \\ p_{H\rightarrow L} & 1 - p_{H\rightarrow L} \end{pmatrix} \quad (4)$$

The two parameters in this Markov process ($p_{L\rightarrow H}$ and $p_{H\rightarrow L}$) were determined using type-II maximum likelihood estimation (MLE).

According to the Bayesian filtering method, the sequence of the subject's cognitive states was estimated based on the sequence of actions ($a_{1:T-1}^*$) and the observed scenes ($o_{1:T}^*$). The posterior probability of the cognitive state at time $t+1$ was obtained from the previous one at time $t$ using the following incremental Bayesian equation:

$$P(c_{h,1:t+1}, h_{1:t+1}, m_{1:t+1}|a_{1:t}^*, o_{1:t+1}^*)$$
$$= \frac{P(c_{h,t+1}, h_{t+1}, m_{t+1}|c_{h,1:t}, h_{1:t}, m_{1:t}, a_{1:t}^*, o_{1:t+1}^*) P(a_t^*|c_{h,1:t}, h_{1:t}, m_{1:t}, a_{1:t-1}^*) P(c_{h,1:t}, h_{1:t}, m_{1:t}|a_{1:t-1}^*, o_{1:t}^*)}{P(a_t^*|a_{1:t-1}^*)} \quad (5)$$

where we used simplified time-series representations like $a_{1:t}^* = \{a_1^*, a_2^*, \dots, a_t^*\}$. By repeating this calculation from first time step, 0, to the terminal time step, $T$, we obtained the posterior probability of the sequence of cognitive states. One likelihood term (the second term in the numerator of Eq. (5)) corresponded to steps 2 and 3 above. Another likelihood term (the first term in the numerator of Eq. (5)) was calculated as follows, based on the subject's behavioral model and corresponding to steps 4 and 5:

$$P(c_{h,t+1}, h_{t+1}, m_{t+1}|c_{h,1:t}, h_{1:t}, m_{1:t}, a_{1:t}^*, o_{1:t+1}^*)$$
$$= P(c_{h,t+1}|c_{h,t}, m_{t+1}) P(h_{t+1}|h_{1:t}, m_{1:t+1}, a_{1:t}^*, o_{1:t+1}^*) P(m_{t+1}|h_t, a_t^*, o_{t+1}^*) \quad (6)$$

When the operant state was update mode ($m_{t+1} = 0$), the second term in Eq. (6) was equal to the previous term:

$$P\left(h_{t+1}|h_{1:t}, m_{t+1} = 0, m_{1:t}, a_{1:t}^*, o_{1:t+1}^*\right) = P(h_t|h_{1:t-1}, m_{1:t}, a_{1:t-1}^*, o_{1:t}^*) \quad (7)$$

because $h_{t+1}$ was determined by $h_t$ and $a_t^*$, without any ambiguity, given $m_{t+1} = 0$. When the operant state was backtrack mode ($m_{t+1} = 1$), a new state inference ($h_{t+1}$) was selected from the re-estimated set of states ($H_{t+1}$) with equal probability (see step 5(b)). $H_{t+1}$ was constructed so that each element was consistent with the history of past $n$-step observed scenes ($o_{t-n+1:t}^*$) and actions ($a_{t-n:t}^*$), and $n$ was subject-wisely estimated by MLE.

There were some exceptional cases that applied to step 5. When the subjects moved to a grid space that had been visited before, but their predicted scene matched the observed scene, $\hat{o} = o^*$, the operant state was set as backtrack mode (action-backtrack mode) because they were considered to have performed inefficient or erroneous exploration. If all the passable doors led to grids that had been explored in such an action-backtrack mode, it resulted in another exceptional case that was regarded as update mode. Note that these exceptional cases were addressed for logical consistency but rarely occurred.

When validating our HMM-based behavioral model, we used the agreement between the model's predicted action ($\hat{a}_t$) and the actual action ($a_t^*$) taken by the subjects. The action was predicted by

$$\hat{a}_t = \underset{a}{\mathrm{argmax}}\, P\left(a|a_{1:t-1}^*, o_{1:t}^*\right) = \underset{a}{\mathrm{argmax}} \sum_{c_{h,t}} \sum_{h_t} \sum_{m_{1:t}} P(a|c_{h,t}, h_t, m_{1:t}, a_{1:t-1}^*)$$
$$P\left(c_{h,t}, h_t, m_{1:t}|a_{1:t-1}^*, o_{1:t}^*\right) \quad (8)$$

which can be calculated as a by-product of Eq. (5). When Eq. (8) provided multiple equally probable actions, we regarded the set of those actions as predicted actions.

When performing model-based analysis, we also used the state confidence, objectively estimated as

$$\hat{c}_{h,t} = \begin{cases} 0 & \text{if } \max_{h_t} P\left(c_{h,t} = 1, h_t\right) \leq \sum_{h_t} P\left(c_{h,t} = 0, h_t\right) \text{ for } h_t \in H \\ 1 & \text{otherwise} \end{cases} \quad (9)$$

where $P\left(c_{h,t}, h_t\right)$ can be obtained by marginalizing Eq. (6) at time step $t$ with respect to $m_t$, where $H$ is the subset of $H_t$ consisting of $h_t$ for which the $P(h_t)$ is maximal in $h_t \in H_t$. The number of elements in $H$ was sometimes greater than one.

We estimated the model parameters for each subject by minimizing the negative log evidence (Supplementary Table 3):

$$\text{Negative log evidence} = -\log \prod_{g=1}^{G} p(A_g^*|\boldsymbol{\theta}) \quad (10)$$

$$p\left(A_g^*|\boldsymbol{\theta}\right) = p(a_1^*|\boldsymbol{\theta}) \prod_{t=1}^{T-1} p(a_{t+1}^*|a_t^*, \boldsymbol{\theta}) \quad (11)$$

Here, $G$ is the number of games, $A_g^*$ is the sequence of actions $a_{1:T-1}^*$, where $T - 1$ is the number of action trials in the game $g$ and $T$ is the number of observations. The set of model parameters is denoted by $\boldsymbol{\theta}$. Note that Eq. (11) is the product of the denominator of Eq. (5) and was obtained by repeating the incremental Bayesian estimation (Eq. (5)). The minimized negative log evidence was also used for the Bayesian model selection (see Supplementary Table 3).

**Statistics and reproducibility.** Imaging data were analyzed using SPM12 (Wellcome Department of Cognitive Neurology, London, UK). The decoding analysis method was implemented using Brain Decoder Toolbox[77], and the searchlight analysis method was implemented using the Decoding Toolbox[78]. Statistical analyses were performed using MATLAB R2017a (Mathworks, Natick, Massachusetts, US). We analyzed the effects of the prediction correctness and the confidence level on the scene choice reaction time with the $R$ package "ARTool"[81,82]. For the behavioral analyses, we used the data of 27 subjects whose scene prediction accuracy in the experimental task was significantly higher than chance (one-sided $z$-test, $p < 0.01$, see also Supplementary Fig. 2). The imaging and decoding analyses included 26 subjects because one subject was excluded due to his/her larger head motion. The decoding accuracies were validated using LOSO or LOGO CV procedures and compared to the theoretical chance level using Wilcoxon signed-rank test. The permutation tests were also applied for the scene prediction and confidence decoding analyses (Supplementary Fig. 4d, e). In the conditional decoding analyses, we applied Wilcoxon rank-sum test to evaluate the difference in the scene prediction decodability between the subsets of data (correct vs incorrect, high vs low scene-prediction-confidence, or high vs low state confidence). Note that, as we simply compared two groups of data, no multiple comparison correction was necessary. Values are expressed as the mean ± SD.

**Reporting summary**. Further information on research design is available in the Nature Research Reporting Summary linked to this article.

## Data availability

The source data underlying the main figures are provided as Supplementary Data 1. All data supporting the main findings are also available via the open-source repository Zenodo[83].

## Code availability

Codes for the computational model of subjects' exploration behavior are available via the open-source repository Zenodo[83].

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

## Acknowledgements

This study was supported by KAKENHI, No. 17H06310, and JP17H06314 from MEXT, Japan and 19H04180 from JSPS, Japan. The authors thank B. Seymour for invaluable comments to improve this study. The authors would also like to thank Editage (www.editage.com) for English language editing.

## Author contributions

W.Y. and S.I. conceived the project; R.K., W.Y., and S.I. designed the research; R.K. and W.Y. performed the research; R.K. analyzed the data; R.K. wrote the draft; and R.K., W.Y., and S.I. prepared the final manuscript.

## Competing interests

The authors declare no competing interests.
