## [Peer Review File · Communications Biology]

Reviewers' comments:

Reviewer #1 (Remarks to the Author):

In their manuscript, Katayama et al. report the results of an fMRI study that aimed at investigating neural representations of scene predictions and of confidence about the predicted scenes. The authors used a maze exploration task in which participants navigated through "rooms" of the maze and predicted the configuration of open and closed doors in these rooms. Using MVPA on BOLD signals during the prediction phase, they decoded correctly predicted room configurations in parietal (SPL, SMG) and frontal (PMd) cortex as well as how confident (high vs. low) the participants were about their predictions in these ROIs and in anterior PFC. The authors conclude that parietal and frontal regions may provide dissociable functions in subjective predictions and confidence.

This is well-written paper that describes an interesting and well-conducted and -analyzed study. The results are believable and the authors' interpretations are plausible. I have some comments on some theoretical and methodological aspects that I hope help the authors to further strengthen their manuscript.

--

1. While I liked the thorough and careful interpretation of the results in the different ROIs in the discussion, I missed a more precise description of the hypotheses and/or tested theories in the introduction. The authors only "predicted that distinct regions of prefrontal and parietal cortices [...] would encode predictions, confidence, and/or interaction between confidence and predictions", which I find quite vague. Did the authors have more specific hypothesis? If not, this may be fine, but then they should explicitly state that the study was more exploratory.

2. The authors based the MVPA on ROIs identified in a quite general univariate contrast. I am concerned that this approach does not reflect the real picture of the representation of scene prediction and confidence in the brain. Are the 4 tested ROIs the only regions in which scene prediction and confidence can be decoded or are there other ROIs (perhaps showing even stronger decoding)? The ROI analysis should be complemented by a searchlight analysis.

3. I found the GLM somewhat underspecified (or not sufficiently explained): Did the authors really just used a single predictor (delay period of prediction trials)? Why not modelling also the action trials as well as the different phases of the prediction trials (confidence choice, scene choice, feedback..) separately? Did the authors include head motion regressors?

4. Why did the authors use sparse logic regression for MVPA? I found this a rather uncommon choice that could be justified in more detail. More common classifiers in fMRI-based MVPA are LDA or SVM, and I am wondering how these classifiers would perform compared with SLR.

5. I understand that the authors used the raw BOLD signal during the delay period for MVPA to ensure that the classifier did not rely on information during the feedback phase (in which the correct scene was shown). First, the authors could explain this choice more explicitly. Second, it is likely that decoding is strongest for BOLD signals 4-6s after a certain event, which means that restricting the analysis to the 4-6s of the delay phase actually excludes valuable information. This view appears to be supported by the increasing decoding in the time-series MVPA (strongest for the last period). To elucidate this issue, the authors could extend the time-series MVPA to the whole prediction trial. This might even provide some interesting additional insights: E.g. the authors could compare the decoding during the delay period (prediction) with the decoding during the scene feedback, and thereby disentangle representations of predicted and perceived scenes.

Minor:

1. Some details about the main MVPA are unclear: "voxel activity patterns during the delay period were used to decode both scene prediction and confidence" Did the authors use the average of the

BOLD signals across the 4-6 volumes?

2. line 211-213: "To examine the sensitivity of the decoders to pattern activation time courses, we employed four consecutive scans starting at five different time points; from the onset of the delay period (first period) to 4 s after the onset (fifth period)." I had a hard time understanding what precisely the authors did and why. Perhaps the authors can explain this in more detail?

3. I had to search in the methods what decoding period (1st to 5th) refers to. I would find it much easier if the authors use time (or volume) instead of # decoding period (in the text and in Fig. 3d,e).

4. lines 198-199: "we performed a univariate general linear model analysis during the prediction of the upcoming scene (first 4 s of the delay period) and found significantly higher brain (BOLD) responses in.."  the contrast is unclear (delay vs. all remaining time points if I see it correctly; but see my comment on the GLM above).

5. line 535: "If an action was not taken within the allotted time, one of the passable doors was chosen by the computer." How often did this happen?

6. I am not sure about the identification of the inferior parietal cluster as SMG. In the right hemisphere, the cluster seems too posterior for SMG. However, I might be wrong – did the authors use an objective method to localize the cluster (e.g. Talairach client)? Alternatively, the authors could label the ROI as inferior parietal lobe (IPL).

7. Figure 2e: I found the circles and triangles a bit hard to distinguish. The authors could consider using separate plots for correct and incorrect predicted scene choices or use different markers/colors.

8. some typos: line 83: decoding-making  decision-making; line 389: electroencephalogram  electroencephalography?

Reviewer #2 (Remarks to the Author):

Katayama and colleagues explored the neural functional patterns associated with predicting the next scene in a maze during uncertainty and proposed a mathematical model to explain the observed behavioural data and to estimate the hidden confidence level of being in a certain state. In general, the paper is concise and well-structured, the experiment is well-described, and I think that the authors used a smart and sophisticated experimental design to investigate scene prediction. Also, the description of the mathematical model is clear, even for non-experts, and it gives strength to the study. However, I found some of the sections confusing and complicated to understand, and important methodological information is missing. I won't comment on the novelty of the study, as I feel I do not have the expertise to judge that. I have three major concerns and a few suggestions for improving the manuscript.

Major points

1. I am quite concerned about the univariate analysis that was used for ROI selection. The description of this analysis is very short and unclear (lines 585 to 595). The design matrix is not clear: it is said that a boxcar function was used to explain sustained activity in the first 4 seconds of the Delay period, however, it seems that no convolution with a hemodynamic model has been computed. Can you please clarify this point? How can you consider the sluggishness of the BOLD signal? And related, are 4 seconds sufficient to capture the shape of the BOLD signal? What about movement parameters and other predictors? Were the trials modelled as different predictors? If the representation changes as a function of the number of prediction trials, I think this should be taken into account. It looks like a big bunch of information is not described here, but it is essential to know it to understand and replicate the analysis.

2. I am a bit worried about the unbalanced design. The Authors say that they deal with the

unbalanced occurrence of the scenes using under-sampling methods for the decoding analysis (line 622), but I am worried about ROI selection. Some of the scenes are more frequent than others (Figure 1c), so these can be estimated better than the ones that are less frequent because there is less noise. In other words, there might be voxels that have been selected because of noise associated with the less frequent scenes. This might be problematic, I am afraid. Please, see also Kriegeskorte et al. (2009, *Nature Neuroscience*, pp 536) and related supplementary materials (pp 1 and 2). I am not sure this is actually a problem, but I would like the Authors to reflect on this and show why this is not a problem. Also, related, what about differences between correct vs incorrect trials? And differences in high vs low confidence? Could these cause issues for the decoding analysis? Also in this case, if you have unbalanced trials, ROI selection might be in favour of one condition or the other and this might explain why the results are in favour of correct trials.

3. The HMM model is generally explained well, but how its results relate to the neural data need to be better explained. First, it took me some mental effort to understand that there are two types of confidence, one that is related to the estimation of the (latent) current state (state confidence) and the other (visible) one that is related to the prediction of the next state (prediction confidence). I think you should make it clearer how these two are different and why they produce different results (Figure 3e and 4e).

Suggestions for improvements

4. I am not sure I understand what the major claim is. The title says: "Decoding scene prediction associated with the dynamically changing confidence during partially-observable maze exploration", which seems too generic for me. I would suggest using a title that already describes the main finding.

5. Related to point 2, at the end of the introduction (lines 87 to 91), the Authors say: "Based on Bayesian models of belief updating in the brain, we predicted that distinct regions of prefrontal and parietal cortices—areas where activity has been frequently observed in spatial navigation and planning (Rodriguez, 2010; Sherrill et al., 2015; Spiers and Maguire, 2007; Viard et al., 2011) — would encode predictions, confidence, and/or interaction between confidence and predictions." This also sounds too generic to me, and it seems obvious that areas in the brain would encode those types of cognitive functions. This might make readers unsatisfied and might also complicate the interpretation of the results. If Authors have predictions about specific areas and their roles, they should state them here, along with the reasons related to those predictions.

6. The labels used for the ROIs are a bit confusing. In Figure 3, the ROIs were called aPFC, PMd, SMG, and SPL. First, in supplementary Table 2, different labels were used, and I would suggest using the same labels throughout the manuscript. Note also that I think there was a typo, as the PMd is missing. Second, it is not clear whether the left and right ROIs for SPL and SMG were combined in the analysis, and if yes, why this is the case.

7. Related to the ROIs, I read that the number of voxels is quite different between regions. Could please the Authors justify the appropriateness of the ROI sizes, and why it is not a problem if they are so different from each other?

8. I would suggest showing the results of the univariate analysis (i.e., betas) also for each ROI in bar plots, including individual data points and by keeping the conditions of interest (correct vs incorrect, high vs low confidence) separated. It is important to show that the decoding observed is not due to differences between the two conditions at the univariate level. This is also somehow related to point 2.

9. Why was the Sparse Logistic Regression used for the decoding analysis? I am not suggesting that this is not appropriate, but I am not familiar with this method, and I think that the Author should justify why they decided to use this method instead of others (e.g., support vector machine, SVM) and explain how the method works. I am saying this because my understanding is that SLR perform a feature selection at the beginning of the analysis, but it is not clear how this affects the dimension of an ROI. How many features are removed at this stage? Would this be a

problem for small ROIs?

10. I think it would help to show a picture of the time course of the ROIs for the group average or for one representative participant to have an idea of the shape of the BOLD signal during the Delay period, to have an immediate grasp of the events in the design and possibly to provide a schematic representation of the 5 windows used for the decoding analysis. I would keep the two conditions, correct and incorrect, separated.

11. Why were five time windows used? And why the decoding was conducted on time-course data and not estimated betas?

12. Why is the maze in Figure 1a 3x3? Is it just an example? If so, it should be clearly stated in the caption, as this might be confusing, as my understanding is that the maze used in the experiment was 5x5.

13. Why certain scenes are less frequent than others? And why scene 3 in particular? Is there an intrinsic reason related to the way the maze is made?

14. Was the maze the same for all participants?

15. How were data in Figure 2b, c, and d aggregated? Are these all participants?

16. Regarding the analysis in Figure 3d,e, shouldn't these results be corrected for multiple comparisons? Was any correction applied? The same applies to other analyses.

17. I am confused about the results shown in Figure 3b. The plot suggests that it is possible to discriminate between scenes in three ROIs. I assume this plot includes both correct and incorrect trials. But wouldn't this be problematic for the classifier to learn the scene pattern, since for the incorrect trials, by definition, the pattern should be not "good" (because it was noisy and thus could not be used by the participant to infer the correct next trial)? And this represents the 1st decoding period shown in Figure 2d, right? Here, the correct vs incorrect responses were kept separated, and the decoding for the 1st and 2nd period for SPL in the incorrect responses is significant. However, if SPL represents the "predicted scene itself" (line 440), it should not be possible to decode the scene when this is not predicted correctly by the participants. And why there is significant decoding only so early? Could the Authors comment on these points?

18. Could the Authors clarify whether a smoothing of 8mm has been used also for the decoding analysis (line 588)? Or were the data left unsmoothed for the decoding analysis? Please, clarify.

Brown: Reviewer's comments

Black: Our replies

Blue: Extracts from the manuscripts

Responses for Reviewer 1

For Major comments:

1. While I liked the thorough and careful interpretation of the results in the different ROIs in the discussion, I missed a more precise description of the hypotheses and/or tested theories in the introduction. The authors only "predicted that distinct regions of prefrontal and parietal cortices [...] would encode predictions, confidence, and/or interaction between confidence and predictions", which I find quite vague. Did the authors have more specific hypothesis? If not, this may be fine, but then they should explicitly state that the study was more exploratory.

We thank the reviewer for the positive comments and constructive reviews. We agree that the previous description of our hypotheses was somewhat vague, so we have now rewritten the corresponding part to be more specific (line 85-).

"We hypothesized that scene prediction in a partially observable environment would be encoded in the parietal and prefrontal cortices, where activity is frequently observed in spatial navigation and planning¹⁶⁻¹⁹, while its confidence would be encoded in the anterior part of the prefrontal cortex^{8,9}. However, it remains unclear whether prediction and confidence interact at the level of neural activity. We supposed that if the confidence modulates the neural representation encoding scene prediction, the decodability of the prediction would vary with the confidence level."

2. The authors based the MVPA on ROIs identified in a quite general univariate contrast. I am concerned that this approach does not reflect the real picture of the representation of scene prediction and confidence in the brain. Are the 4 tested ROIs the only regions in which scene prediction and confidence can be decoded or are there other ROIs (perhaps showing even stronger decoding)? The ROI analysis should be complemented by a searchlight analysis.

We appreciate this important suggestion. In this study, we focused on investigating the relationship between the neural representation of scene prediction and its confidence, rather than fully exploring the location of the cortical areas encoding the metacognition involved in prediction. Thus, we first defined task-related brain regions (ROIs) using an unsupervised ROI extraction method, and then examined the label-related decodability in each task-related ROI.

However, agreeing that searchlight analysis can legitimately be added as a complement to the ROI-based analysis, we have now performed a whole-brain searchlight analysis with a 10 mm radius sphere centered around a given voxel for the session-wise unsmoothed beta estimates. To create subject-level whole-brain accuracy maps, we used a linear support vector machine [1] and the accuracy of each voxel was evaluated using leave-one-session-out cross-validation. The individual accuracy maps were normalized and smoothed using a Gaussian kernel (8mm FWHM) and then applied to the group random effect analysis using anatomically-localized cerebral cortex.

[1] Hebart, M. N., Gorgen, K. & Haynes, J. D. The decoding toolbox (TDT): A versatile software package for multivariate analyses of functional imaging data. *Front. Neuroinform.* **8**, 88 (2015).

In the scene prediction searchlight analysis, the seven different scenes predicted by the subjects were modeled in the GLM as seven individual regressors during the delay period. Other than the delay period, the same regressors as in the original GLM were used. The results showed that the bilateral SPL and IPL were the only clusters that showed significantly higher decoding accuracy than chance level (voxel level $p < 0.001$; cluster level, FWE corrected $p < 0.05$), and these regions overlapped well with the ROIs shown in the original univariate analysis. No clusters were found in aPFC, and this result was also consistent with the result of our ROI-based scene prediction decoding analysis.

The results of the whole-brain searchlight analysis for the scene prediction.

For the confidence level, we conducted a searchlight analysis with GLM using two regressors according to the subject's confidence level (high or low) in the delay period. The results showed that the regions in which confidence can be decoded are widely distributed in the cerebrum. This result is quite different from that of a related study (Morales et al. [2]), which demonstrated that confidence during perceptual and memory tasks was decodable in the aPFC and fronto-

parietal midline. This difference may have arisen because their searchlight-based decoder used brain activity when the subjects *reported* their confidence level, whereas our decoder used brain activity when the subjects were instructed to predict the upcoming scene, and the subjects did not actively make a conscious confidence judgement.

We have added the results of the searchlight analyses as a supplement to our univariate and unsupervised ROI analysis in the Method section (line 648-).

“To complement the univariate ROI analysis, we also performed whole-brain searchlight analyses (see Supplementary Methods). For scene prediction, we confirmed that there were no clusters other than the four ROIs extracted by the univariate analysis that showed significantly higher decoding accuracy than chance (voxel level, $p < 0.001$; cluster level, FWE-corrected $p < 0.05$). For confidence level, we found that the regions where the confidence can be decoded were widely distributed over the cerebrum, which may be attributed to the task design in which subjects were instructed to predict the upcoming scene during the delay period.”

[2] Morales, J., Lau, H. & Fleming, S. M. Domain-general and domain-specific patterns of activity supporting metacognition in human prefrontal cortex. *J. Neurosci.* **38**, 3534–3546 (2018).

3. I found the GLM somewhat underspecified (or not sufficiently explained): Did the authors really just use a single predictor (delay period of prediction trials)? Why not modelling also the action trials as well as the different phases of the prediction trials (confidence choice, scene choice, feedback..) separately? Did the authors include head motion regressors?

We apologize that the previous descriptions of the GLM were indeed insufficient. For each session, we modeled seven events as the regressors in the GLM: action selection, and moving scenery in the action trials, and delay period, confidence evaluation, feedback for confidence evaluation, predicted scene choice and scene choice feedback in the prediction trials. Additionally, six motion correction parameters estimated during realignment were included as nuisance variables for the GLM. We have now rewritten the descriptions of the GLM as follows (line 632-):

“Our univariate analysis was based on the generalized linear model (GLM) approach. Our GLM included seven regressors coding for onsets and durations of events in each session: action selection and moving scenery in the action trials, delay period, confidence evaluation, feedback for confidence evaluation, predicted scene choice, and scene choice feedback in the prediction trials.

The durations of choice-related events (action selection, confidence evaluation, and predicted scene choice) were defined as the time between the option presentation and the subject's response. The durations of moving and choice feedback were fixed to 1.5 s. For the delay period (regressor-of-interest), although the time length varied trial-by-trial, we modeled it as a boxcar function for 4sec (the minimum duration of the delay period). These regressors were convolved with a hemodynamic response function (HRF). Additionally, motion correction parameters produced during realignment were included as nuisance variables for the GLM. The first-level GLM analysis was performed using the contrast vector whose element was 1 for the regressor-of-interests, and 0 otherwise. We then performed a group random effect analysis using anatomically localized cerebral cortex to find cortical voxels that were significantly and commonly activated during prediction across all subjects. We established statistical thresholds at the voxel level of $p < 0.001$ (uncorrected) and at the cluster level of $p < 0.05$. We extracted regions of interest (ROIs) from the identified voxels, and the BOLD signal patterns in each ROI were used for the decoding analysis."

4. Why did the authors use sparse logic regression for MVPA? I found this a rather uncommon choice that could be justified in more detail. More common classifiers in fMRI-based MVPA are LDA or SVM, and I am wondering how these classifiers would perform compared with SLR.

In our decoding analyses, the number of features (the number of voxels in each ROI) was about 2.5~13 times the sample size, and this high dimensionality required a powerful dimensionality reduction method to build robust predictive models. Here, we applied SLR because this algorithm incorporates Bayesian automatic selection of relevant features (voxels), which is called automatic relevance determination (ARD), thereby preventing overfitting. This method has been used for MVPA in a number of previous studies [1,2,3]. As the reviewer suggested, we tested the performance of SVM and LDA (please see the figure below), and found that SVM showed significantly lower accuracy than the others. Although LDA decoders were as good as SLR in our case, LDA can cause "data piling" and show a bad generalization property especially when applied to high-dimensional but small sample-sized data [4]. To clarify the reason for our usage of SLR, the following statement was added to the Methods (line 677-):

"We used a sparse logistic regression (SLR)⁷⁶ as a supervised learning algorithm because it is able to automatically select relevant features (voxels), thereby preventing overfitting problems in high-dimensional neuroimaging data."

[1] Shikauchi, Y. & Ishii, S. Decoding the view expectation during learned maze navigation from human fronto-parietal network. *Scientific Reports* vol. 5 1–13 (2015).

[2] Cortese, A., Amano, K., Koizumi, A., Kawato, M. & Lau, H. Multivoxel neurofeedback selectively modulates confidence without changing perceptual performance. *Nat. Commun.* **7**, (2016).

[3] Horikawa, T. & Kamitani, Y. Generic decoding of seen and imagined objects using hierarchical visual features. *Nat. Commun.* **8**, (2017).

[4] Qiao, Z., Zhou, L. & Huang, J. Effective Linear Discriminant Analysis for High Dimensional, Low Sample Size Data. *Proc. World Congr.* **II**, 3–8 (2008).

Decoding accuracies using different classification methods (SLR, SVM, LDA).

To evaluate the SLR decoder's performance, we compared it with those by SVM and LDA. For both scene prediction (a) and confidence (b) decoders, we used the voxel activity patterns of the 6th decoding period. This period was used because of its increasing decodability in our time-series analysis. The decoder's accuracy was evaluated using leave-one-session-out cross-validation. Each box extends from the lower to upper quartiles, with a horizontal line at the median. The whiskers show 1.5 IQR, and cross markers indicate the outliers. Significance was tested using a one-sided Wilcoxon signed rank test compared to chance (dotted line) (*: $p < 0.05$, **: $p < 0.01$, ***: $p < 0.001$). In order not to increase in the number of figures, this figure is not included in the current manuscript.

- I understand that the authors used the raw BOLD signal during the delay period for MVPA to ensure that the classifier did not rely on information during the feedback phase (in which the correct scene was shown). First, the authors could explain this choice more explicitly. Second, it is likely that decoding is strongest for BOLD signals 4-6s after a certain event, which means that restricting the analysis to the 4-6s of the delay phase actually excludes valuable information. This view appears to be supported by the increasing decoding in the time-series MVPA (strongest for the last period). To elucidate this issue, the authors could extend the time-series MVPA to the whole prediction trial. This might even provide some interesting additional

insights: E.g. the authors could compare the decoding during the delay period (prediction) with the decoding during the scene feedback, and thereby disentangle representations of predicted and perceived scenes.

We appreciate this interesting suggestion. In the previous manuscript, we restricted the time-series MVPA up to the 5th decoding period, i.e., usage of the voxel activity patterns of 4~7s since the onset of delay period, to avoid the influence from the subsequent visual stimuli. However, we agree that extending the MVPA time-series might provide additional insights, and so we have now examined the decoding accuracies when extended to the whole prediction trial. Here the result of the t -th period means when the decoder used the BOLD signal from t s to $t+3$ s since the delay onset. Please note that this new definition of decoding period is different from the one in the previous manuscript. The onset of each event since the delay onset was as follows: confidence choice, 5.0s; confidence feedback, 6.0s; scene choice, 7.5s; scene feedback, 9.1s; next scene presentation, 10.6s (average of all subjects and all prediction trials).

We added the following figure as the Supplementary Figure (line 362-):

Supplementary Figure 4. Design of the time-series decoding analysis and supplementary results.

a) The design of the time-series decoding analysis and the temporal relationship of events in the prediction trial. In the time-series decoding analyses, the decoding result for the t -th decoding period was for the BOLD signal from t s to $t+3$ s (i.e., $(t+1)$ -th to $(t+4)$ -th volumes) after the onset of the delay period. The onset of each event since the delay onset (D) was: confidence choice (CC), 5.0s; confidence choice feedback (CF), 6.0s; scene choice, 7.5s; scene choice feedback (SF), 9.1s; next scene presentation (SP), 10.6s (mean for all prediction trials of 26 subjects included in the decoding analyses). Because the BOLD signal increases 4–6 s after its evoking event, the effect of the scene choice period, during which the subjects' predicted (chosen) scene is displayed, was not taken into account in our decoding analysis.

b,c) Time-series decoding results. Each panel shows the decoding accuracies of the scene prediction (**b**) and the subjects' reported confidence level about the scene prediction (**c**) within each ROI. The decoding accuracies were evaluated using leave-one-session-out (LOSO) cross validation (CV). The solid black lines represent the median, gray shaded areas indicate the range between the upper and lower quartiles, and the dotted lines indicate the range of $1.5 \times IQR$. Cross-markers indicate the outliers. Significance was tested using a one-sided

Wilcoxon signed-rank test compared to chance (dashed line). Unfilled markers indicate significant thresholds (circle: $p < 0.05$, square: $p < 0.01$, diamond: $p < 0.001$).

When decoding the scene prediction, the accuracy gradually increased from the 5th to 7th periods, which included the BOLD signals after 4–6sec from the delay onset, in SPL and IPL. However, there was neither a clear peak nor a sharp decrease in accuracy after the delay period. This could be due to the wide range of time during which the upcoming scene was predicted, or the need to maintain the predicted scene throughout the prediction trial to report it in the subsequent scene choice event. The scene prediction was also decodable in aPFC but only after the delay period (i.e., the 9th period) during which the subjects were required to make a choice for the scene prediction. This is now discussed in the Discussion section (line 434-).

“Using a decoding analysis for four ROIs that were identified by univariate analysis, we showed that both scene prediction and confidence level were decodable from the voxel activity patterns in the SPL, IPL, and PMd. The time-series analysis also showed that the scene prediction decoding accuracy gradually increased from the 5th to 7th periods, which included the BOLD signals after 4–6s from the delay onset in the SPL and IPL. Although there was no clear peak or sharp drop in accuracy following the delay period, this could be due to the wide range of time during which the upcoming scene was predicted, or to the need to maintain the predicted scene throughout the prediction trial to report it in the subsequent scene choice period. Meanwhile, our decoding analysis demonstrated that only confidence could be decoded from the aPFC activity.”

When decoding the confidence, the accuracy tended to increase from the 6th to 8th periods in SPL, IPL and PMd, but no such tendency was observed in aPFC. The confidence decodability in SPL, IPL and PMd could have peaked in the 6th to 8th periods, because these areas are involved in the generation process of prediction which would be affected by the confidence (or the uncertainty), whereas the neural activities in the ROIs can be affected by the information of subsequent visual stimuli (scene choice period and scene presentation in the following action trial) rather than prediction itself. On the other hand, the confidence decodability was sustained in aPFC, possibly because it was necessary to maintain the confidence information for the state estimation update and the action selection in the following trials.

Here, we examined the accuracy over the entire prediction trial. However, since our study focuses on the brain activity involved in the prediction itself, and in order to avoid the situation in which the accuracy of the decoder is affected by the information of the subsequent visual stimuli (visual cues in the scene choice period and scene presentation in the following action

trial), we present the results of the time-series decoding analyses up to the 8th decoding period in the manuscript.

We added the justification for choosing nine decoding periods for the time-series MVPA in the Method section (line 661-).

“In the time-series decoding analysis, the decoder at each time t in the time course (t -th decoding period) used as its input the voxel-wise BOLD signal intensities averaged over four volumes corresponding to t s to $t+3$ s (i.e., $(t+1)$ -th to $(t+4)$ -th scan volumes) after the onset of the delay period (Supplementary Figure 4a). We limited the time-series decoding analysis up to the 8th period in order to cover 4–6 s after the delay onset, which exhibited the peak brain activity evoked by the delay cue, and to ensure that the scene prediction decoder was as unaffected as possible by information provided in the scene choice period (average 7.5 s after the delay onset), during which the subjects’ predicted (chosen) scene was displayed.”

Accordingly, we modified Figure 3, which shows the results of the time-series conditional decoding analyses as follows. For the representative data of scene prediction and confidence decoding (Fig. 3bc), we used the 6th decoding period to show the results because of its increasing decodability in the time-series analysis (the 0th period was used in the previous manuscript).

Figure 3. Imaging analysis results and decoding accuracies.

b,c) Decoding accuracies for scene prediction (**b**, six types of scenes) and its confidence level (**c**, high or low) within each ROI evaluated using leave-one-session-out (LOSO) cross-validation (CV). Each box extends from the lower to upper quartiles, with a horizontal line at the median. The whiskers represent $1.5 \times \text{IQR}$, and cross markers indicate the outliers.

Significance was tested using a one-sided Wilcoxon signed rank test compared to chance (dashed line) (**: $p < 0.01$, ***: $p < 0.001$). These figures represent the results of the scene prediction and confidence decoders using the 6th decoding period as representative data because of its increasing decodability in our time-series analysis; the overall results of the time-series decoding analysis are shown in **Supplementary Figure 4b-e**.

d,e) Time-series scene prediction decoding results within each ROI when the data were categorized binarily according to the prediction correctness (**d**, correct vs. incorrect trials) and the confidence level (**e**, high-confidence vs. low-confidence trials) of the prediction trial. For example, 'correct' indicates the accuracy of the scene prediction decoder trained and tested with only the trials in which subjects' upcoming scene selections were correct (correct-only decoder). The decoding accuracies were evaluated using the leave-one-game-out (LOGO) CV. The solid lines reflect the median, the shaded areas indicate the range between the upper and lower quartiles, and the dotted lines indicate the range of $1.5 \times \text{IQR}$. The cross-markers indicate outliers. Significance was tested using a one-sided Wilcoxon signed-rank test (unfilled circle: $p < 0.05$, unfilled square: $p < 0.01$, unfilled diamond: $p < 0.001$) compared to the chance level (dashed line). The color of the horizontal

line below the plots reflects a significant difference between the two categories of trials in each decoding period (one-sided Wilcoxon rank sum test, light gray: $p < 0.05$, dim gray: $p < 0.01$, black: $p < 0.001$).

For Minor comments:

1. Some details about the main MVPA are unclear: "voxel activity patterns during the delay period were used to decode both scene prediction and confidence" Did the authors use the average of the BOLD signals across the 4-6 volumes?

We performed a time-series decoding analysis of each decoder using the average BOLD signal across four volumes starting from different time points. Please refer to the reply to major comment #5 for the details.

2. line 211-213: "To examine the sensitivity of the decoders to pattern activation time courses, we employed four consecutive scans starting at five different time points; from the onset of the delay period (first period) to 4 s after the onset (fifth period)." I had a hard time understanding what precisely the authors did and why. Perhaps the authors can explain this in more detail?

We apologize for our previous insufficient explanation. We have rewritten the description of the time-course analysis as follows (line 233-):

"To probe the sensitivity of the decoders to pattern activation time course, we constructed the decoders of the scene prediction and the subject's reported confidence level at nine different time points, the 0th to the 8th decoding periods: the decoders at the t -th period used four consecutive scans starting from t s after the onset of the delay period (Supplementary Figure 4a)."

Please also refer to the reply to major comment #5.

3. I had to search in the methods what decoding period (1st to 5th) refers to. I would find it much easier if the authors use time (or volume) instead of # decoding period (in the text and in Fig. 3d,e).

Thank you for giving us this suggestion. We have changed the way we describe the decoding periods to correspond to the time from the delay period onset. Please refer to the reply to major comment #5 above for the new descriptions.

4. lines 198-199: "we performed a univariate general linear model analysis during the prediction of the upcoming scene (first 4 s of the delay period) and found significantly higher brain (BOLD) responses in.."  the contrast is unclear (delay vs. all remaining time points if I see it correctly; but see my comment on the GLM above).

We apologize for the lack of explanation in the previous manuscript. We have added the following sentence to the Method section (line 641):

"The first-level GLM analysis was performed using the contrast vector whose element was 1 for the regressor-of-interests, and 0 otherwise."

Please also refer to the reply to major comment #3.

5. line 535: "If an action was not taken within the allotted time, one of the passable doors was chosen by the computer." How often did this happen?

The number of unresponsive action trials (no action within the 2.5s allotted time) was 12.7 ± 9.8 ($2.8 \pm 2.0\%$) for 27 subjects who were included in the behavioral analyses. Also, the subjects chose impassable doors for 2.7 ± 3.1 trials ($0.6 \pm 0.7\%$). Those data are now added to the Method section (line 574-).

"If subjects chose an impassable door, they remained in the same state and the same scene was presented again in the subsequent trial (2.7 ± 3.1 trials for 27 subjects, $0.6 \pm 0.7\%$). If an action was not taken within the allotted time, one of the passable doors was chosen by the computer (12.7 ± 9.8 trials, $2.8 \pm 2.0\%$)."

6. I am not sure about the identification of the inferior parietal cluster as SMG. In the right hemisphere, the cluster seems too posterior for SMG. However, I might be wrong – did the authors use an objective method to localize the cluster (e.g. Talairach client)? Alternatively, the authors could label the ROI as inferior parietal lobe (IPL).

We appreciate this important suggestion. Based on Talairach Atlas Labels (on xjview), we confirmed that the peak voxel in the left inferior parietal cluster was located in SMG, whereas that in the right inferior parietal cluster was in IPL, slightly posterior to SMG, as the reviewer suggested. We rewrote the corresponding labels as IPL.

7. Figure 2e: I found the circles and triangles a bit hard to distinguish. The authors could consider using separate plots for correct and incorrect predicted scene choices or use different markers/colors.

We now depict the boxplots with different colors between subplots for the correct and incorrect choices.

8. some typos: line 83: decoding-making  decision-making; line 389: electroencephalogram  electroencephalography?

Thanks. We fixed these typos.

Brown: Reviewer's comments

Black: Our replies

Blue: Extracts from the manuscripts

Responses for Reviewer 2

For Major comments:

1. I am quite concerned about the univariate analysis that was used for ROI selection. The description of this analysis is very short and unclear (lines 585 to 595). The design matrix is not clear: it is said that a boxcar function was used to explain sustained activity in the first 4 seconds of the Delay period, however, it seems that no convolution with a hemodynamic model has been computed. Can you please clarify this point? How can you consider the sluggishness of the BOLD signal? And related, are 4 seconds sufficient to capture the shape of the BOLD signal? What about movement parameters and other predictors? Were the trials modelled as different predictors? If the representation changes as a function of the number of prediction trials, I think this should be taken into account. It looks like a big bunch of information is not described here, but it is essential to know it to understand and replicate the analysis.

We apologize for the lack of adequate information about the univariate analysis for the ROI selection in the previous manuscript. For each session, we modeled seven events as the regressors in the GLM: action selection and moving scenery in the action trials, and delay period (regressor of interest), confidence evaluation, feedback for confidence evaluation, predicted scene choice and scene choice feedback in the prediction trials. These regressors were convolved with a hemodynamic response function (HRF). Since the length of the delay period of the prediction trials varied from trial to trial (4–6s) and was unknown for the subjects, prediction of the upcoming scene would have been done in 4 seconds. We thus considered that the 4 seconds boxcar function is sufficient to capture the brain activity related to the scene prediction. Additionally, six motion correction parameters estimated during realignment were included as nuisance variables for the GLM.

This point was also made by Reviewer 1, and we have rewritten the descriptions of the GLM (Please see the reply to the major comment #3 of Reviewer 1).

We did not include the number of prediction trials as a regressor, because we thought that the neuronal representation of scene prediction would not change depending on the number of trials. Just to be sure, however, we examined the GLM results when the number of prediction

trials was added as an additional regressor, and confirmed that there was no major difference in the univariate analysis results.

The results of the univariate analysis including the effect of the number of trials.

The statistical thresholds at the voxel level of $p < 0.001$ (uncorrected) and at the cluster level of $p < 0.05$ (FWE-corrected).

2. I am a bit worried about the unbalanced design. The Authors say that they deal with the unbalanced occurrence of the scenes using under-sampling methods for the decoding analysis (line 622), but I am worried about ROI selection. Some of the scenes are more frequent than others (Figure 1c), so these can be estimated better than the ones that are less frequent because there is less noise. In other words, there might be voxels that have been selected because of noise associated with the less frequent scenes. This might be problematic, I am afraid. Please, see also Kriegeskorte et al. (2009, Nature Neuroscience, pp 536) and related supplementary materials (pp 1 and 2). I am not sure this is actually a problem, but I would like the Authors to reflect on this and show why this is not a problem. Also, related, what about differences between correct vs incorrect trials? And differences in high vs low confidence? Could these cause issues for the decoding analysis? Also in this case, if you have unbalanced trials, ROI selection might be in favour of one condition or the other and this might explain why the results are in favour of correct trials.

This is an important point, and we agree that unbalanced samples could cause a small distortion due to a possible noise from the less frequent samples as Kriegeskorte et al. suggested [1]. To address this issue, we performed three additional analyses, as detailed below, the results of which did not suggest any fundamental bias or distortion in the results of the study.

First, we looked at the correlation between the number of samples (frequency) and the decoding accuracy of each scene to see if noise caused lower decoding accuracy for scenes with lower

frequency. There was no significant positive correlation in all four ROIs at 9 decoding timings (i.e., 36 decoders in total; note that, as the response to major comment #5 of Reviewer 1, we increased the number of decoding periods in the current manuscript). We added these results to the Result as to refer to the Supplementary Table 2.

In the Result section (line 259-):

“There was no significant positive correlation between the number of samples (frequency) and the decoding accuracy of each scene in all four ROIs at nine decoding periods (i.e., 36 decoders in total; Supplementary Table 2). Therefore, the unbalanced number of samples was found not to distort the decoding analyses.”

In the Supplementary Tables (line 512-):

Supplementary Table 2. The correlation between the number of samples and the decoding accuracy for each scene.

	SPL	IPL	PMd	aPFC
0th	$r=-9.1\times 10^{-2}, p=0.26$	$r=-0.11, p=0.8$	$r=-0.14, p=7.4\times 10^{-2}$	$r=-5.2\times 10^{-2}, p=0.52$
1st	$r=-0.15, p=7.0\times 10^{-2}$	$r=-7.3\times 10^{-2}, p=0.37$	$r=-0.13, p=0.10$	$r=-4.1\times 10^{-2}, p=0.61$
2nd	$r=-0.13, p=0.11$	$r=-0.15, p=5.6\times 10^{-2}$	$r=-0.19, p=2.1\times 10^{-2}$	$r=-9.6\times 10^{-2}, p=0.23$
3rd	$r=-0.11, p=0.16$	$r=-7.6\times 10^{-2}, p=0.34$	$r=-0.13, p=0.10$	$r=-0.14, p=7.1\times 10^{-2}$
4th	$r=-5.4\times 10^{-2}, p=0.51$	$r=-3.8\times 10^{-2}, p=0.64$	$r=-0.16, p=5.1\times 10^{-2}$	$r=-3.2\times 10^{-2}, p=0.69$
5th	$r=-7.2\times 10^{-2}, p=0.37$	$r=-4.3\times 10^{-2}, p=0.59$	$r=-9.9\times 10^{-2}, p=0.22$	$r=-2.6\times 10^{-2}, p=0.74$
6th	$r=-3.4\times 10^{-2}, p=0.67$	$r=-6.2\times 10^{-2}, p=0.44$	$r=-5.6\times 10^{-2}, p=0.49$	$r=-4.9\times 10^{-2}, p=0.55$
7th	$r=-8.9\times 10^{-2}, p=0.27$	$r=-0.13, p=0.12$	$r=-0.16, p=4.3\times 10^{-2}$	$r=-2.9\times 10^{-2}, p=0.72$
8th	$r=-0.11, p=0.18$	$r=-0.19, p=1.5\times 10^{-2}$	$r=-0.18, p=2.5\times 10^{-2}$	$r=-7.6\times 10^{-2}, p=0.35$

Second, to ensure that certain infrequent scenes did not affect the ROI selection, we performed univariate analyses for the trials when the subjects were predicting all but one of the seven scenes (i.e., 7 times in total), and then performed the conjunction analysis over them. In the conjunction analysis, we used 7 contrast maps of all 26 subjects generated in the above univariate analysis to localize the brain voxels which were activated commonly regardless of

the predicted scene types. This conjunction analysis revealed that the four ROIs found in the original univariate analysis were similarly extracted even when one scene was removed out of the seven.

The result of the conjunction analysis.

The statistical thresholds at the voxel level of $p < 0.001$ (uncorrected) and at the cluster level of $p < 0.05$ (FWE-corrected).

The above results suggested that there was no substantial bias or distortion in the results of this study. However, Kriegeskorte et al. also point out that the decoding accuracy may be positively biased if the data used for ROI selection is also used for decoder evaluation. To examine if this had arisen in our study, we additionally investigated the decoders' accuracies using leave-one-session-out (LOSO) cross-validation (CV) for ROI selection: in each fold of CV, one session of data was excluded for later decoder validation, and the remaining sessions were used for ROI selection and decoder training. Please note that this CV procedure avoids the classic double-dipping problem in the field of statistics. The results showed no consistent bias in the decoding accuracy, i.e., there was no improvement in the accuracy when decoding test data was also used for ROI selection.

The decoding results compared between the different ROI selection methods.

We evaluated the scene prediction (a) and confidence (b) decoders' accuracies with the leave-one-session-out cross-validation. In each fold, we also performed ROI selection, so that there was no information leak when testing the validation data. For both types of decoders, we used the voxel activity patterns of the 6th decoding period. Each box extends from the lower to upper quartiles, with a horizontal line at the median. The whiskers show 1.5 IQR, and cross markers indicate the outliers. Significance was tested using a one-sided Wilcoxon signed rank test compared to chance (dotted line) (*: $p < 0.05$, **: $p < 0.01$, ***: $p < 0.001$). To avoid the increase in the figure number, we do not include the above figure in the current manuscript.

The reviewer was also concerned that only certain conditions might support ROI selection; however, the proportions of correct and incorrect trials were 54.6% and 45.4%, and the proportions of high and low confidence were 57.3% and 42.8%, and there was no sample size bias in both cases (two-sided Wilcoxon signed rank test, correct vs incorrect, $p = 0.12$; high vs low confidence, $p = 0.15$). We thus consider that it is unlikely that these conditions had caused the ROI selection bias.

[1] Kriegeskorte, N., Simmons, W. K., Bellgowan, P. S. & Baker, C. I. Circular analysis in systems neuroscience: The dangers of double dipping. *Nat. Neurosci.* **12**, 535–540 (2009).

3. The HMM model is generally explained well, but how its results relate to the neural data need to be better explained. First, it took me some mental effort to understand that there are two types of confidence, one that is related to the estimation of the (latent) current state (state confidence) and the other (visible) one that is related to the prediction of the next state

(prediction confidence). I think you should make it clearer how these two are different and why they produce different results (Figure 3e and 4e).

We thank the reviewer for this suggestion. According to our view of incremental Bayesian filtering processes, scene prediction should incorporate a mental simulation process utilizing the memory of the maze structure, after inferring the current state in the maze (please see equation (5) in the Supplementary Methods (line 231-), which expresses the difference between the scene prediction and the state inference). Since the scene prediction follows the state inference, it includes additional uncertainty that would vary from subject to subject, which may reflect the incompleteness of the maze structure memory. In fact, the concordance between the subject-reported scene prediction confidence and the model-estimated confidence of the state inference was about 60%, suggesting that there may be a difference between these two kinds of confidence.

Corresponding to this difference, we found some difference in the time-course decodability between SPL and IPL. Interestingly, for the scene prediction decoder using the SPL, the accuracy profile was different for the high- and low-scene-confidence trials, whereas for the decoder using the IPL, the accuracy profile was different for the high- and low-state-confidence trials. In addition, the scene prediction decodability in the high-state-confidence trials with the IPL responses reached a peak a little earlier (6th period) than that in the high-scene-confidence trials with the SPL responses (7th period).

We clarified the difference between the scene prediction confidence and the state confidence and explained in more detail the reason why these two types of confidence lead to different decoding results in the revised Discussion as follows (line 522-):

“According to our view of an incremental Bayesian filtering process, scene prediction should incorporate a mental simulation process utilizing the memory of the maze structure after inferring the current state in the maze. Since the scene prediction follows the state inference, it includes additional uncertainties that vary from subject to subject, such as the degree of imperfectness of the maze structure memory. The subject-reported scene-prediction-confidence may not necessarily match the estimated (but objectively reconstructed by our model) state-confidence, as represented in the subjects’ brain. In fact, the agreement between the scene-prediction-confidence (subject-reported, high or low) and the state-confidence (model-estimated, high or low) was $63.2 \pm 10.5\%$ for all prediction trials, suggesting that there may be a difference between these two types of

confidence. The difference in the time-course decodability between the SPL and IPL corresponded to this difference between the confidence types.”

Responses for Minor comments (Suggestions for improvements):

4. I am not sure I understand what the major claim is. The title says: “Decoding scene prediction associated with the dynamically changing confidence during partially-observable maze exploration”, which seems too generic for me. I would suggest using a title that already describes the main finding.

We have changed the title so as to reflect our main findings more clearly, as follows:

“Confidence modulates the decodability of scene prediction during partially-observable maze exploration”

5. Related to point 2, at the end of the introduction (lines 87 to 91), the Authors say: “Based on Bayesian models of belief updating in the brain, we predicted that distinct regions of prefrontal and parietal cortices—areas where activity has been frequently observed in spatial navigation and planning (Rodriguez, 2010; Sherrill et al., 2015; Spiers and Maguire, 2007; Viard et al., 2011)—would encode predictions, confidence, and/or interaction between confidence and predictions.” This also sounds too generic to me, and it seems obvious that areas in the brain would encode those types of cognitive functions. This might make readers unsatisfied and might also complicate the interpretation of the results. If Authors have predictions about specific areas and their roles, they should state them here, along with the reasons related to those predictions.

We apologize that our motivation and hypotheses were described rather vaguely in the previous manuscript. In response to this comment (and also the major comment #1 of Reviewer 1), we have rewritten the corresponding part to be more specific (line 84-). Please refer our reply to the major comment #1 of Reviewer 1.

6. The labels used for the ROIs are a bit confusing. In Figure 3, the ROIs were called aPFC, PMd, SMG, and SPL. First, in supplementary Table 2, different labels were used, and I would suggest using the same labels throughout the manuscript. Note also that I think there was a typo, as the PMd is missing. Second, it is not clear whether the left and right ROIs for SPL and SMG were combined in the analysis, and if yes, why this is the case.

Thank you for this suggestion. We have standardized the labels throughout the manuscript including the figure captions and the supplementary materials. Note that we now use 'IPL' instead of 'SMG' based on the anatomical definition (please see the reply to the minor comment #6 of Reviewer 1).

In our decoding analyses, the left and right ROIs for SPL and IPL were combined. We used the combined ROIs because previous neuroimaging studies on spatial tasks showed that the SPL is activated bilaterally [1,2,3], and recent decoding studies on prediction in spatial navigation tasks also used combined bilateral ROIs for MVPA [4,5]. We additionally performed MVPAs using the left or right SPL and IPL separately and confirmed that the decoding accuracies were significantly higher than chance, as well as the combined ROIs. In the current manuscript, we have clarified that the ROIs for SPL and IPL were used bilaterally as follows (line 229-):

“We performed a multi-voxel pattern analysis using the voxel-wise activity patterns of the SPL, IPL, PMd, and aPFC. For the SPL and IPL, the left and right ROIs were combined, as in the recent decoding studies on navigation^{1,2}.”

[1] Seydell-Greenwald, A., Ferrara, K., Chambers, C. E., Newport, E. L. & Landau, B. Bilateral parietal activations for complex visual-spatial functions: Evidence from a visual-spatial construction task. *Neuropsychologia* **106**, 194 (2017).

[2] Peer, M., Ron, Y., Monsa, R. & Arzy, S. Processing of different spatial scales in the human brain. *Elife* **8**, (2019).

[3] Stilla, R., Deshpande, G., LaConte, S., Hu, X. & Sathian, K. Posteromedial Parietal Cortical Activity and Inputs Predict Tactile Spatial Acuity. *J. Neurosci.* **27**, 11091–11102 (2007).

[4] Elliott Wimmer, G. & Büchel, C. Learning of distant state predictions by the orbitofrontal cortex in humans. *Nat. Commun.* **10**, (2019).

[5] Shikauchi, Y. & Ishii, S. Decoding the view expectation during learned maze navigation from human fronto-parietal network. *Scientific Reports* vol. 5 1–13 (2015).

Decoding accuracies compared bilateral and unilateral ROIs.

The decoding accuracies of scene prediction (a) and confidence level (b) within the left, right, or bilateral (the left and right ROIs were combined) SPL and IPL. For both kinds of decoders, we used the 6th decoding period, and the decoder's accuracy was evaluated using leave-one-session-out (LOSO) cross-validation (CV). Each box extends from the lower to upper quartiles, with a horizontal line at the median. The whiskers show 1.5 IQR, and cross markers indicate the outliers. Significance was tested using a one-sided Wilcoxon signed rank test compared to chance (dotted line) (*: $p < 0.05$, **: $p < 0.01$, ***: $p < 0.001$). To avoid an excessive increase in the number of figures, we do not include the above figure in the current manuscript.

7. Related to the ROIs, I read that the number of voxels is quite different between regions. Could please the Authors justify the appropriateness of the ROI sizes, and why it is not a problem if they are so different from each other?

In our study, we used sparse logistic regression (SLR) [1]. Yamashita et al. reported that if the number of input features is much larger than that of relevant features (e.g., when the number of input features is more than 100 and the number of relevant features is 10), the classification accuracy and the number of features selected by the SLR are almost independent of the initial dimensionality. In this study, 40-50 features (voxels) were extracted as relevant features by the scene prediction decoders and 15-20 features by the confidence decoders (please see the figure attached to our reply to your comment #9 below). While the size of ROI ranged from 497 to 2686 voxels, the number of selected features was fairly consistent. Furthermore, there was no significant correlation between the number of selected voxels or ROI size and the decoding accuracies of both scene prediction and confidence. Thus, the difference in ROI size does not seem to seriously bias in the results.

[1] Yamashita, O., Sato, M. A., Yoshioka, T., Tong, F. & Kamitani, Y. Sparse estimation automatically selects voxels relevant for the decoding of fMRI activity patterns. *Neuroimage* **42**, 1414–1429 (2008).

8. I would suggest showing the results of the univariate analysis (i.e., betas) also for each ROI in bar plots, including individual data points and by keeping the conditions of interest (correct vs incorrect, high vs low confidence) separated. It is important to show that the decoding observed is not due to differences between the two conditions at the univariate level. This is also somehow related to point 2.

Thank you for this suggestion. We confirmed that the beta values in three ROIs (SPL, IPL and PMd) were not significantly different between the compared conditions (correct vs incorrect, high vs low scene confidence, or high vs low state confidence). We have clarified these results in the revised Result section and have added the figure below to the Supplementary Materials.

In the Results section (line 289-):

“We also confirmed that the beta estimates and the percent signal changes (PSCs) in the ROIs (SPL, IPL, and PMd) were not significantly different between the compared conditions (Supplementary Figure 3d). Therefore, the differences in the scene prediction decoding accuracy were not due to the activity differences that were examined in the univariate analyses.”

Supplementary Figure 3: Supplementary results of the univariate-level ROI analyses.

d,e) The beta value of each ROI in which the scene prediction was decodable, in comparison between correct and incorrect trials (d), and high- and low-scene-confidence trials (e). Each plot displays the data of the subjects included in the corresponding conditional decoding analyses. There was no significant difference between the paired conditions for all three ROIs (two-sided Wilcoxon rank sum test).

9. Why was the Sparse Logistic Regression used for the decoding analysis? I am not suggesting that this is not appropriate, but I am not familiar with this method, and I think that the Author should justify why they decided to use this method instead of others (e.g., support vector machine, SVM) and explain how the method works. I am saying this because my understanding is that SLR perform a feature selection at the beginning of the analysis, but it is not clear how this affects the dimension of an ROI. How many features are removed at this stage? Would this be a problem for small ROIs?

In our decoding analyses, the number of features (the number of voxels in each ROI) was approximately 2.5-13 times the sample size, and this high dimensional nature required a powerful dimensionality reduction method to build a robust predictive model. Here, we applied SLR because this algorithm utilizes Bayesian automatic selection of relevant features (voxels), called automatic relevance determination (ARD), thereby preventing overfitting. This method has been used for MVPA in previous studies [1,2,3]. We also tested the performance of SVM as suggested by the reviewer, but SVM showed significantly lower accuracy than SLR. In the current manuscript, we clarified the reason why we used SLR. Please refer to the reply to the major comment #4 of Reviewer 1, which is a similar question.

The number of selected features was about 40-50 (voxels) for the scene prediction decoder and 15-20 for the confidence decoder (median). While the size of ROI varied from 497 to 2686 voxels, the number of selected features was almost constant between the different ROIs.

The number of relevant features selected by the SLR decoders.

Each panel shows the number of features selected by the scene prediction decoders (a) and the subject's reported confidence level decoders (b) using SLR. Each box extends from the lower to upper quartiles, with a horizontal line at the median. The whiskers show 1.5 IQR, and cross markers indicate the outliers.

[1] Shikauchi, Y. & Ishii, S. Decoding the view expectation during learned maze navigation from human fronto-parietal network. *Scientific Reports* vol. 5 1–13 (2015).

[2] Cortese, A., Amano, K., Koizumi, A., Kawato, M. & Lau, H. Multivoxel neurofeedback selectively modulates confidence without changing perceptual performance. *Nat. Commun.* **7**, (2016).

[3] Horikawa, T. & Kamitani, Y. Generic decoding of seen and imagined objects using hierarchical visual features. *Nat. Commun.* **8**, (2017).

10. I think it would help to show a picture of the time course of the ROIs for the group average or for one representative participant to have an idea of the shape of the BOLD signal during the Delay period, to have an immediate grasp of the events in the design and possibly to provide a schematic representation of the 5 windows used for the decoding analysis. I would keep the two conditions, correct and incorrect, separated.

We appreciate this interesting suggestion. Accordingly, we now additionally show the percent signal changes of the ROIs during the prediction trials as the Supplementary Figure (see below).

Supplementary Figure 3: Supplementary results of the univariate-level ROI analyses.

a,b,c) Time-series of the percent signal change of each ROI during the prediction trials **(a)**, and when the data were categorized into two according to the prediction correctness **(b)** and the confidence level **(c)**. The neural responses were sampled at each TR (every 1 s) from the onset of the delay period (marked as 0 s). The lines and the colored area indicate the average and SEM over the 26 subjects included in the imaging and the decoding analyses **(a)**, or the subjects included in the corresponding conditional decoding analyses **(b,c)**. The color of the horizontal line below the plots **(b,c)** indicates the significant difference between the two categories of trials in each timepoint (two-sided Wilcoxon rank sum test, light gray: $p < 0.05$, dim gray: $p < 0.01$, black: $p < 0.001$).

We also added a schematic representation of the decoding periods used for the time-series decoding analyses as Supplementary Figure 4a. Please refer to the reply to the major comment #5 of Reviewer 1.

11. Why were five time windows used? And why the decoding was conducted on time-course data and not estimated betas?

In the previous manuscript, we used five time-windows for the decoding analyses, covering the whole delay period, but excluding the effect of visual stimuli provided after the delay period. In the current manuscript, we extended the time-series MVPA to 9 decoding periods to cover the whole prediction trial (please see the reply to the major comment #5 of Reviewer 1).

We appreciate that MVPA can be done with time-course data (linear-detrended and block-wise averaged) or estimated betas as its decoder input. Etzel et al. examined the decoding accuracy with these two different decoder input settings and showed that the results varied depending on the type of datasets [1]. Here we used time-course data because this input setting performed better than the other setting (i.e., using estimated betas) in continuous sampling, event-related experiments [1] such as in our case.

[1] Etzel, J. A., Valchev, N. & Keysers, C. The impact of certain methodological choices on multivariate analysis of fMRI data with support vector machines. *Neuroimage* **54**, 1159–1167 (2011).

12. Why is the maze in Figure 1a 3x3? Is it just an example? If so, it should be clearly stated in the caption, as this might be confusing, as my understanding is that the maze used in the experiment was 5x5.

Thank you for pointing this out. We used the 3x3 maze in Figure 1a for explaining the task. We clarified this by adding the following sentence to the caption of Figure 1a (line 100-):

“The 3×3 maze in this figure is used to explain the task, and the maze used in the actual experiment was of a 5×5 size (Supplementary Figure S1a).”

13. Why certain scenes are less frequent than others? And why scene 3 in particular? Is there an intrinsic reason related to the way the maze is made?

In this study, we created the maze to satisfy the following two constraints: 1) the structure can be learned within the limited training time and memory capacity, and 2) at least two consecutive observations from the initial state are required to identify the current state in order to well define the partial-observability of the maze. We also took into account that the frequency of scenes in the maze was not biased, and the distribution was actually not different from the uniform distribution (Supplementary Figure 1b). Although scene #3 rarely occurred as a scene predicted by the subjects, it probably happened by chance. We have added the constraints for the maze design in the Methods section as follow (line 565-):

“We created a maze to satisfy partial observability, that is, at least two consecutive observations from the initial state are required to identify the current state.”

14. Was the maze the same for all participants?

We used the same maze so that there would be no difference in the level of difficulty between individuals. We now clarify this in the Method section (line 562-):

“We used the same maze for all subjects so that there would be no difference in the level of task difficulty between individuals.”

15. How were data in Figure 2b, c, and d aggregated? Are these all participants?

The data shown in Figure 2b-d was of 27 subjects included in the behavioral analyses. We clarify this in the caption of Figure 2 (line 135-):

“Note that the data shown in (b-f) was of 27 subjects included in the behavioral analyses.”

16. Regarding the analysis in Figure 3d,e, shouldn't these results be corrected for multiple comparisons? Was any correction applied? The same applies to other analyses.

Figure 3d and 3e compare the accuracies of two scene-prediction decoders which were trained and tested by different subsets of data: correct vs incorrect (Figure 3d) and high vs low scene-confidence (Figure 3e). As we simply compared two groups of data, no multiple comparison correction was necessary; the same applies to Figure 4e.

17. I am confused about the results shown in Figure 3b. The plot suggests that it is possible to discriminate between scenes in three ROIs. I assume this plot includes both correct and incorrect trials. But wouldn't this be problematic for the classifier to learn the scene pattern, since for the incorrect trials, by definition, the pattern should be not "good" (because it was noisy and thus could not be used by the participant to infer the correct next trial)? And this represents the 1st decoding period shown in Figure 2d, right? Here, the correct vs incorrect responses were kept separated, and the decoding for the 1st and 2nd period for SPL in the incorrect responses is significant. However, if SPL represents the "predicted scene itself" (line 440), it should not be possible to decode the scene when this is not predicted correctly by the participants. And why there is significant decoding only so early? Could the Authors comment on these points?

We apologize for the confusing explanation in the previous manuscript. In our analyses, the decoding target (label) was the scene chosen by the subject *as the predicted scene, not the true upcoming scene*, regardless of whether the prediction was correct or not. It is possible to decode the predicted scene even on the incorrect trials (see also Shikauchi et al. [1]), but as the reviewer pointed out, the data are often noisy and therefore likely to be less accurate in decoding than on the correct trials (Fig. 3d). We have added a note on this point in the revised Results section (line 241-).

"In the following decoding analyses, we used the scenes chosen by the subjects for the target labels of the scene decoder regardless of whether they were correct or not."

In the very early periods, the decoding accuracy based on SPL activity was significantly higher than chance in both correct and incorrect trials. This may reflect the predictions that occur immediately after the previous action trial. However, it is difficult to examine the reason why the accuracy subsequently decreased in the incorrect trials and increased in the correct trial under the current experimental design.

[1] Shikauchi, Y. & Ishii, S. Decoding the view expectation during learned maze navigation from human fronto-parietal network. *Scientific Reports* vol. 5 1–13 (2015).

18. Could the Authors clarify whether a smoothing of 8mm has been used also for the decoding analysis (line 588)? Or were the data left unsmoothed for the decoding analysis? Please, clarify.

We have added the following description to the revised Method section (line 656-):

“Voxel activity patterns during the delay period were used to decode both scene prediction and confidence. All fMRI data were spatially realigned, normalized, and smoothed with a Gaussian kernel (8 mm FWHM), and preprocessed with linear trend removal and z-score normalization for each voxel in every run over the time-series, but not convolved with HRF.”

REVIEWERS' COMMENTS:

Reviewer #2 (Remarks to the Author):

The Authors made a great effort to respond to the comments of both Reviewers. I must say that all major comments were carefully addressed and I cannot see further major concerns.

Though, I would suggest the Authors to add in the manuscript the response to Reviewer2 - Major Point #2 regarding ROI selection. I see they added the information about the no correlation between the number of samples and the decoding accuracy in the Result section, but I think Readers would also appreciate the LOSO CV analysis. I also suggest adding either in the manuscript or in the supplemental materials the figures and comments about the number of features selected by the SLR decoder and how there is consistency across ROIs.

Finally, although you explain well why you decided to use the SLR for decoding in the rebuttal letter, I think you should spend a few more words in the manuscript (see Reviewer1 - Major Point #4): why is it the right method to use here? You can say that it is a Bayesian method and has been used in previous studies.

Minor point:

In your response to Reviewer1 - Major Point #2, you wrote that the individual accuracy maps were smoothed using 8mm Gaussian kernel. Is this appropriate? Can you cite other studies doing this? The searchlight analysis already provides a sort of smoothing of the maps, so I wonder if it's ok to smooth again.

Brown: Reviewer's comments

Black: Our replies

Blue: Extracts from the manuscripts

Responses for Reviewer 2

For comments:

1. Though, I would suggest the Authors to add in the manuscript the response to Reviewer2 - Major Point #2 regarding ROI selection. I see they added the information about the no correlation between the number of samples and the decoding accuracy in the Result section, but I think Readers would also appreciate the LOSO CV analysis. I also suggest adding either in the manuscript or in the supplemental materials the figures and comments about the number of features selected by the SLR decoder and how there is consistency across ROIs.

Thank you for the suggestion. We added the results of the LOSO CV analysis, showing that there was no improvement in the decoding accuracy when the test data was also used for the ROI selection. We also added the results about the number of features selected by the SLR decoders.

In the Results section (line 172):

"Additionally, while the size of ROIs varied from 497 to 2686 voxels, the number of selected features was almost constant between the different ROIs (Supplementary Figure S4f-g). Therefore, the unbalanced number of samples and dimensionalities of ROIs were found not to distort the decoding results. We also confirmed that there was no positive bias in the decoding accuracy even though the data for decoder evaluation was also used for the ROI selection²¹ (Supplementary Figure S4h-i)."

And in the Supplementary Figure (line 97):

Supplementary Figure 4. Design of the time-series decoding analysis and supplementary results.

f,g) Numbers of relevant features selected by the SLR decoders. Each panel shows the numbers of features selected by the scene prediction decoders (**f**) and by the subject-reported confidence level decoders (**g**) using SLR. Each box extends from the lower to upper quartiles, with a horizontal line at the median. The whiskers show $1.5 \times IQR$, and cross markers indicate the outliers.

h,i) Decoding results compared between different ROI selection methods. The accuracy of the scene prediction decoder (**h**) and the confidence decoder (**i**) was evaluated by leave-one-session-out cross-validation. In each fold, ROI selection was performed to ensure that there was no information leak when testing the validation data. For both decoders, the voxel activity patterns of the 6th decoding period was used. Each box extends from the lower to upper quartiles, with a horizontal line at the median. The whiskers show $1.5 \times IQR$, and cross markers indicate the outliers. Significance was tested using a one-sided Wilcoxon signed rank test compared to chance (*: $p < 0.05$, **: $p < 0.01$, ***: $p < 0.001$).

- Finally, although you explain well why you decided to use the SLR for decoding in the rebuttal letter, I think you should spend a few more words in the manuscript (see Reviewer1 - Major Point #4): why is it the right method to use here? You can say that it is a Bayesian method and has been used in previous studies.

We have rewritten the rationale for applying SLR to MVPA (line 602):

"We used a sparse logistic regression (SLR)" as a supervised learning algorithm because it incorporates Bayesian automatic selection of relevant features (voxels), which prevents overfitting problems in high-dimensional neuroimaging data. This method has been used for MVPA in previous studies^{2,12,25}."

For minor comment:

3. In your response to Reviewer1 - Major Point #2, you wrote that the individual accuracy maps were smoothed using 8mm Gaussian kernel. Is this appropriate? Can you cite other studies doing this? The searchlight analysis already provides a sort of smoothing of the maps, so I wonder if it's ok to smooth again.

Thank you for pointing this out. In the searchlight analyses, we smoothed the individual accuracy maps using an 8mm Gaussian kernel for the group-level searchlight analyses, similar to the methods used in some previous studies [1-2]. We now cited these studies in the Supplementary methods section.

[1] Morales, Jorge, Hakwan Lau, and Stephen M. Fleming. "Domain-general and domain-specific patterns of activity supporting metacognition in human prefrontal cortex." *Journal of Neuroscience* 38.14 (2018): 3534-3546.

[2] Momennejad, Ida, and John-Dylan Haynes. "Encoding of prospective tasks in the human prefrontal cortex under varying task loads." *Journal of Neuroscience* 33.44 (2013): 17342-17349.